# Offline Multi-Agent Reinforcement Learning with Implicit Global-to-Local Value Regularization

**Xiangsen Wang**[1*]   **Haoran Xu**[2*]   **Yinan Zheng**[3]   **Xianyuan Zhan**[3,4†]

[1] Beijing Jiaotong University [2] UT Austin [3] Tsinghua University
[4] Shanghai Artificial Intelligence Laboratory

wangxiangsen@bjtu.edu.cn, haoran.xu@utexas.edu,
zhengyn23@mails.tsinghua.edu.cn, zhanxianyuan@air.tsinghua.edu.cn

## Abstract

Offline reinforcement learning (RL) has received considerable attention in recent years due to its attractive capability of learning policies from offline datasets without environmental interactions. Despite some success in the single-agent setting, offline multi-agent RL (MARL) remains to be a challenge. The large joint state-action space and the coupled multi-agent behaviors pose extra complexities for offline policy optimization. Most existing offline MARL studies simply apply offline data-related regularizations on individual agents, without fully considering the multi-agent system at the global level. In this work, we present OMIGA, a new offline multi-agent RL algorithm with implicit global-to-local value regularization. OMIGA provides a principled framework to convert global-level value regularization into equivalent implicit local value regularizations and simultaneously enables in-sample learning, thus elegantly bridging multi-agent value decomposition and policy learning with offline regularizations. Based on comprehensive experiments on the offline multi-agent MuJoCo and StarCraft II micro-management tasks, we show that OMIGA achieves superior performance over the state-of-the-art offline MARL methods in almost all tasks. Our code is available at `https://github.com/ZhengYinan-AIR/OMIGA`.

## 1   Introduction

Multi-agent reinforcement learning (MARL) is an active research area to tackle many real-world problems that involve multi-agent systems, such as autonomous vehicle coordination[1], network traffic routing [2], and multi-player strategy games [3]. Although MARL has made some impressive progress in recent years, most of its successes are restricted to simulation environments. In most real-world applications, building high-fidelity simulators can be rather costly or even infeasible, and online interaction with the real system during policy learning is also expensive or risky. Inspired by the recently emerged offline RL methods [4, 5, 6, 7, 8, 9, 10], equipping MARL with offline learning capability has become an attractive direction to tackle real-world multi-agent tasks.

Offline RL focuses on learning optimal policies with pre-collected offline datasets with no further environmental interaction. Under the offline setting, evaluating value function on out-of-distribution (OOD) samples can cause extrapolation error accumulation on Q-values, leading to erroneously over-estimated values and misguiding policy learning [4, 5]. This issue, which is also called distributional shift [5], is one of the core challenges in offline RL. Current offline RL methods tackle this issue primarily by incorporating various forms of data-related regularizations to restrict policy learning

---

[*]Work done during the internships with Institute for AI Industry Research (AIR), Tsinghua University.
[†]Corresponding Author.

37th Conference on Neural Information Processing Systems (NeurIPS 2023).

from deviating too much from the behavioral data. However, extending the same offline regularization technique to the multi-agent setting poses some unique challenges. The joint state-action space grows exponentially with the number of agents, making the global-level regularization hard to compute and may also result in very sparse constraints under the huge joint state-action space, especially when the size and coverage of the offline dataset are limited.

To tackle the challenges, some recent works [11, 12, 13] attempt to design offline MARL algorithms heuristically by enforcing local-level regularizations. By using the Centralized Training with Decentralized Execution (CTDE) framework [14] to decompose the global value function into a combination of local value functions, these works turn the original offline multi-agent RL problem into the offline single-agent RL problem and apply policy or value regularizations at the local level. While straightforward and easy to implement, simply enforcing local-level regularization cannot guarantee the induced regularization at the global-level still remains valid, which can potentially lead to over-conservative policy learning. Existing approaches offer no guarantee whether the optimized local policies are jointly optimal under a given value decomposition scheme. Furthermore, because the offline regularization of most existing methods is completely imposed from the local-level without considering the global information, these methods fail to capture coordinated agent behaviors and credit assignment in the multi-agent system.

In this paper, we aim to give a principled approach to the offline MARL problem. We start by imposing global-level regularizations to form a *behavior-regularized* MDP. We then find that using some particular regularization choice in this MDP will have closed-form solutions, which naturally turn the global-level regularizations into local-level ones, under some mild assumptions. Finally, we show that the local-level regularization can be derived in an implicit way by using only samples from the offline datasets, without knowing the behavior policy or approximating the regularization term. We dub this new algorithm as offline multi-agent RL algorithm with implicit global-to-local value regularization (OMIGA). In OMIGA, the local-level value regularization is rigorously derived from the global regularization under value decomposition, thus also capturing global information and the impact of multi-agent credit assignment. Under this design, we can also guarantee that the learned local policies are jointly optimal at the global level. Also, OMIGA enables complete in-sample learning without querying OOD action samples, which enjoys better stability during policy learning.

We evaluate our method using various types of offline datasets on both multi-agent MuJoCo [15] and StarCraft Multi-Agent Challenge (SMAC) tasks [16]. Under all settings, OMIGA achieves better performance and enjoys faster convergence compared with other strong baselines.

## 2   Related Work

**Offline reinforcement learning.** Due to the absence of online data collection, offline policy learning suffers from severe distributional shift and exploitation error accumulation issues when evaluating the value function on OOD actions [4, 5]. As a result, existing offline RL methods adopt various approaches to learn policies pessimistically and regularize policy learning close to data distribution. Policy constraint methods [4, 5, 9, 17, 18, 19] add explicit or implicit policy constraints to regularize the policy from deviating too much from behavioral data. Value regularization methods [20, 21, 22, 23] learn conservatively estimated value functions on OOD data. Uncertainty-based methods add uncertainty penalization terms on value functions [24, 25] or rewards [26, 27, 28] to enable pessimistic policy learning. Recently, in-sample learning methods [7, 8, 10, 29] provide a new direction to avoid the distributional shift by learning value functions and policies completely within data. These methods eliminate the involvement of policy-generated OOD samples during policy evaluation, thus enjoying superior learning stability. Our method embeds multi-agent value decomposition within the in-sample learning paradigm, which fully exploits the multi-agent problem structure for offline policy optimization.

**Multi-agent reinforcement learning.** The complexity of multi-agent RL problems typically arises from the huge joint action space [30]. In recent years, the CTDE framework [31, 32] has been proposed to decouple agents' learning and execution phases to handle the exploding action space issue. Under the CTDE framework, agents are trained in a centralized manner with global information and make decisions with learned local policies during execution. Some representative works are the value decomposition methods [33, 34, 35, 36], which decompose the global Q-value function into a set of local Q-value functions for scalable multi-agent policy optimization.

There have been some recent attempts to design MARL algorithms for the offline setting, which can be broadly categorized as RL-based [11, 12, 13, 37] and goal-conditioned supervised learning (GCSL) based methods [38, 39]. Existing RL-based methods are typically built upon the CTDE framework and perform offline policy regularization at the local level. For example, ICQ [11] uses importance sampling to implicitly constrain local policy learning on OOD samples. OMAR [12] adopts CQL [20] to learn local Q-value functions and adds zeroth-order optimization to avoid bad local optima. In these methods, the multi-agent modeling and offline RL ingredients are fragmented, which cannot guarantee that the local-level regularizations still remain valid at the global level, and also ignores the cooperative behavior among agents. Moreover, there is no consideration of the value decomposition's influence on the optimal local policies learned with offline regularizations. On the other hand, GCSL-based methods [38, 39] leverage the sequential data modeling capability of transformer architecture to solve offline MARL tasks. However, recent studies also show that GCSL-based methods can be over-conservative [8] and cannot guarantee optimality under stochastic environments [40, 41].

In this work, our proposed OMIGA is an RL-based method, which uses an implicit global-to-local value regularization scheme to tackle the aforementioned limitations, and offers an elegant solution to marry multi-agent modeling and offline learning.

## 3 Preliminaries

### 3.1 Notations

We focus on the fully cooperative multi-agent tasks, which can be modeled as a multi-agent Partially Observable Markov Decision Process (POMDP) [42], defined by a tuple $G = \langle \mathcal{S}, \mathcal{A}, \mathcal{P}, r, \mathcal{Z}, \mathcal{O}, n, \gamma \rangle$. $s \in \mathcal{S}$ denotes the true state of the environment. $\mathcal{A}$ is the action set for each of the $n$ agents. At each step, an agent $i \in \{1, 2, ...n\}$ chooses an action $a_i \in \mathcal{A}$, forming a joint action $\boldsymbol{a} = (a_1, a_2, ...a_n) \in \mathcal{A}^n$. $P(\mathbf{s}'|\mathbf{s}, \boldsymbol{a}) : \mathcal{S} \times \mathcal{A}^n \times \mathcal{S} \rightarrow [0, 1]$ is the transition dynamics to the next state $s'$. $\gamma \in [0, 1)$ is a discount factor. In the partial observable environment, each agent receives an observation $o_i \in \mathcal{O}$ at each step based on an observation function $\mathcal{Z}(\mathbf{s}, i) : \mathcal{S} \times N \rightarrow \mathcal{O}$, and we denote $\boldsymbol{o} = (o_1, o_2, ...o_n)$. All agents share the same global reward function $r(\mathbf{o}, \boldsymbol{a}) : \mathcal{O} \times \mathcal{A}^n \rightarrow \mathbb{R}$. In cooperative MARL, all agents aim to learn a set of policies $\pi_{tot} = \{\pi_1, \cdots, \pi_n\}$ that jointly maximize the expected discounted returns $\mathbb{E}_{\boldsymbol{a} \in \pi_{tot}, \mathbf{o} \in \mathcal{O}} \left[ \sum_{t=0}^{\infty} \gamma^t r(\boldsymbol{o}_t, \boldsymbol{a}_t) \right]$. Under the offline setting, a pre-collected dataset $\mathcal{D}$ is obtained by sampling with the behavior policy $\mu_{tot} = \{\mu_1, \cdots, \mu_n\}$ and the policy learning is conducted entirely with the data samples in $\mathcal{D}$ without any environment interactions.

### 3.2 CTDE Framework and Value Decomposition

In MARL, the joint action space increases exponentially with the increase in the number of agents. Therefore, it is difficult to directly query an optimal joint action from the global Q-value function $Q_{tot}(\boldsymbol{o}, \boldsymbol{a})$, and the global Q-value function could be negatively affected by the suboptimality of individual agents. To address these problems, the Centralized Training with Decentralized Execution (CTDE) framework [14, 31, 32] is proposed. In the training phase, agents can access the full environment information and share each other's experiences. In the execution phase, each agent chooses actions only according to its individual observation $o_i$. Through the CTDE framework, optimization at the individual level results in the optimization of the joint action space, which avoids the aforementioned problems. Value decomposition methods [33, 34, 35, 36] are popular solutions under the CTDE framework to achieve the decomposition of the joint action space. Value decomposition is also used in recent offline MARL methods [11, 12], which decomposes the global Q-value function $Q_{tot}(\boldsymbol{o}, \boldsymbol{a})$ into a linear combination of local Q-value functions $Q_i(o_i, a_i)$:

$$Q_{tot}(\boldsymbol{o}, \boldsymbol{a}) = \sum_i w_i(\boldsymbol{o}) Q_i(o_i, a_i) + b(\boldsymbol{o}),$$

$$w_i \geq 0, \ \forall i = 1 \cdots, n \tag{1}$$

where $w_i(\boldsymbol{o})$ and $b(\boldsymbol{o})$ capture the weights and offset on local Q-value functions.

### 3.3 Behavior-Regularized MDP in Offline RL

To avoid the distributional shift in offline RL, existing approaches typically incorporate various data-related regularizations on rewards, value functions, or the policy optimization objective. In our work, we are specifically interested in the treatment that adds a behavior regularizer to the rewards $r(s, a)$ to penalize OOD samples [10]. This framework leads to a special behavior-regularized MDP, which optimizes a policy with the following objective:

$$\max_{\pi} \mathbb{E} \left[ \sum_{t=0}^{\infty} \gamma^t \left( r\left(s_t, a_t\right) - \alpha f\left( \pi\left(a_t | s_t\right), \mu\left(a_t | s_t\right)\right)\right)\right], \tag{2}$$

where $\alpha$ is a scale parameter, and $f(\cdot, \cdot)$ is the function that captures the divergence between $\pi$ and $\mu$, which is similar to the entropy regularization in SAC [43]. In the above objective, we regularize the deviation between the learned policy $\pi$ and behavior policy $\mu$, so as to avoid the distributional shift issue. This behavior-regularized MDP corresponds to the following modified policy evaluation operator $\mathcal{T}_f$:

$$\left(\mathcal{T}_f^{\pi}\right) Q(s, a) := r(s, a) + \gamma \mathbb{E}_{s'|s,a}\left[V\left(s'\right)\right]$$

$$\left(\mathcal{T}_f^{\pi}\right) V(s) := \mathbb{E}_{a \sim \pi}\left[r(s, a) + \gamma \mathbb{E}_{s'|s,a}\left[V\left(s'\right)\right]\right],$$

where

$$V(s) = \mathbb{E}_{a \sim \pi}\left[Q(s, a) - \alpha f\left(\pi\left(a_t | s_t\right), \mu\left(a_t | s_t\right)\right)\right].$$

In the next section, we will derive our proposed method OMIGA based on the behavior-regularized framework, and demonstrate the benefits and desired properties of this framework for the multi-agent setting.

## 4 Method

In this section, we formally present our implicit global-to-local value regularization approach OMIGA for offline MARL and explain how it can be integrated into effective offline learning. We begin with the multi-agent POMDP with a reverse KL global value regularization added to the rewards. We then show how we can make an equivalent reformulation that naturally converts the global-level regularization into local-level value regularizations. Lastly, we derive a practical algorithm that implicitly enables local-level value regularizations via in-sample learning and provide a thorough discussion of it.

### 4.1 Multi-Agent POMDP with Global Value Regularization

When extending Eq.(2) to multi-agent POMDP, we can get the following learning objective:

$$\max_{\pi_{tot}} \mathbb{E} \left[ \sum_{t=0}^{\infty} \gamma^t \left( r\left(\boldsymbol{o}_t, \boldsymbol{a}_t\right) - \alpha f\left( \pi_{tot}\left(\boldsymbol{a}_t | \boldsymbol{o}_t\right), \mu_{tot}\left(\boldsymbol{a}_t | \boldsymbol{o}_t\right)\right)\right)\right]$$

However, unlike the single-agent setting, it is difficult to directly compute the regularization term between the global policy $\pi_{tot}$ and the global behavior policy $\mu_{tot}$ due to the huge state-action space in multi-agent RL. Can we choose a function $f$ to allow certain forms of policy decomposition so as to simplify the calculations?

In this work, we find that choosing the regularization function $f$ to be the reverse KL divergence, which means $f(\pi_{tot}, \mu_{tot}) = \log(\pi_{tot}/\mu_{tot})$, leads to natural decomposition of the global regularization into the summation of a set of local regularizations, by only assuming the factorization structure of the behavior policy (i.e., $\mu_{tot}(\boldsymbol{a}|\boldsymbol{o}) = \prod_{i=1}^{n} \mu_i\left(a_i | o_i\right)$). Note that using reverse KL divergence as the behavior constraint has been widely studied in both online and offline single-agent RL literature [43, 17], and has been shown to produce the nice mode-seeking behavior to avoid OOD actions in the offline setting [18]. Plug in the specific form of $f$, and we obtain the following global policy evaluation operator:

$$\left(\mathcal{T}_f^{\pi_{tot}}\right) Q_{tot}(\boldsymbol{o}, \boldsymbol{a}) := r(\boldsymbol{o}, \boldsymbol{a}) + \gamma \mathbb{E}_{\boldsymbol{o}'|\boldsymbol{o},\boldsymbol{a}}\left[V_{tot}\left(\boldsymbol{o}'\right)\right] \tag{3}$$

$$\left(\mathcal{T}_f^{\pi_{tot}}\right) V_{tot}(\boldsymbol{o}) := \mathbb{E}_{\boldsymbol{a} \sim \pi_{tot}}\left[r(\boldsymbol{o}, \boldsymbol{a}) + \gamma \mathbb{E}_{\boldsymbol{o}'|\boldsymbol{o},\boldsymbol{a}}\left[V_{tot}\left(\boldsymbol{o}'\right)\right]\right], \tag{4}$$

where

$$V_{tot}(\boldsymbol{o}) = \mathbb{E}_{\boldsymbol{a} \sim \pi_{tot}} \left[ Q_{tot}(\boldsymbol{o}, \boldsymbol{a}) - \alpha \log \left( \frac{\pi_{tot}(\boldsymbol{a}|\boldsymbol{o})}{\mu_{tot}(\boldsymbol{a}|\boldsymbol{o})} \right) \right] \tag{5}$$

**Theorem 4.1.** *Define $\mathcal{T}_f^*$ the case where the global policy in $\mathcal{T}_f^{\pi_{tot}}$ is the optimal policy $\pi_{tot}^*$, then $\mathcal{T}_f^*$ is a $\gamma$-contraction.*

The proof is in Appendix A. This theorem indicates global Q-value will converge to the Q-value under the optimal policy $\pi_{tot}^*$ when applying the fixed-point iteration with the policy evaluation operator.

We now aim to analyze and derive the closed-form solution of the optimal value functions $Q_{tot}^*$ and $V_{tot}^*$. According to the Karush-Kuhn-Tucker (KKT) conditions where the derivative of a Lagrangian objective function with respect to the global policy is zero at the optimal solution, we have the following proposition:

**Proposition 4.2.** *For a behavior-regularized multi-agent POMDP with $f(\pi_{tot}, \mu_{tot}) = \log(\pi_{tot}/\mu_{tot})$, the optimal global policy $\pi_{tot}^*$ and its optimal value functions $Q_{tot}^*$ and $V_{tot}^*$ satisfy the following optimality condition:*

$$Q_{tot}^*(\boldsymbol{o}, \boldsymbol{a}) = r(\boldsymbol{o}, \boldsymbol{a}) + \gamma \mathbb{E}_{\boldsymbol{o}'|\boldsymbol{o}, \boldsymbol{a}} \left[ V_{tot}^*(\boldsymbol{o}') \right]$$
$$V_{tot}^*(\boldsymbol{o}) = u^*(\boldsymbol{o}) + \alpha \tag{6}$$
$$\pi_{tot}^*(\boldsymbol{a}|\boldsymbol{o}) = \mu_{tot}(\boldsymbol{a}|\boldsymbol{o}) \cdot \exp \left( \frac{Q_{tot}^*(\boldsymbol{o}, \boldsymbol{a}) - u^*(\boldsymbol{o})}{\alpha} - 1 \right)$$

*where $u(\boldsymbol{o})$ is a normalization term and has a optimal value $u^*$ that makes the corresponding optimal policy $\pi_{tot}^*$ satisfy $\sum_{\boldsymbol{a} \in \mathcal{A}^n} \pi_{tot}^*(\boldsymbol{a}|\boldsymbol{o}) = 1$.*

The proof is in Appendix A. Note that Proposition 4.2 can be further simplified. Since $V_{tot}^*(\boldsymbol{o}) = u^*(\boldsymbol{o}) + \alpha$, $u^*$ and $V_{tot}^*$ can be converted to each other without any approximation. For the global policy, replacing $u^*(\boldsymbol{o})$ with $V_{tot}^*(\boldsymbol{o}) - \alpha$, we get the following formulation:

$$\pi_{tot}^*(\boldsymbol{a}|\boldsymbol{o}) = \mu_{tot}(\boldsymbol{a}|\boldsymbol{o}) \cdot \exp \left( \frac{Q_{tot}^*(\boldsymbol{o}, \boldsymbol{a}) - V_{tot}^*(\boldsymbol{o})}{\alpha} \right) \tag{7}$$

Now we have the relationship among the optimal global policy $\pi_{tot}^*$, behavior policy $\mu_{tot}$, Q-value function $Q_{tot}^*$ and state-value function $V_{tot}^*$. Under the multi-agent setting, due to the exponential growth of the joint state-action space with the number of agents, these global values are hard to be evaluated. We now show that we can resort to the value decomposition strategy to further derive a tractable relationship between local policies and value functions.

## 4.2 Global-to-Local Value and Policy Decomposition

In this work, we introduce the following value decomposition scheme for both the global Q-value function and the state-value function:

$$Q_{tot}(\boldsymbol{o}, \boldsymbol{a}) = \sum_i w_i(\boldsymbol{o}) Q_i(o_i, a_i) + b(\boldsymbol{o})$$
$$V_{tot}(\boldsymbol{o}) = \sum_i w_i(\boldsymbol{o}) V_i(o_i) + b(\boldsymbol{o}) \tag{8}$$
$$w_i \geq 0, \ \forall i = 1 \cdots, n$$

Compared with existing offline MARL methods [11, 12, 13] that only decompose the global Q-value function $Q_{tot}$, we additionally decompose the global state-value function $V_{tot}$. The decomposition of $Q_{tot}$ and $V_{tot}$ share a common weight function $w_i(\boldsymbol{o})$, since the credit assignment on $Q$ and $V$ should be related. It should also be noted that $V_{tot}$ is free of the joint action space and thus not affected by OOD actions under offline learning.

If we incorporate the value decomposition scheme Eq. (8) into the optimal global policy $\pi_{tot}^*$ in Eq. (7) and utilize the property of the exponential function, we can naturally decompose the optimal global policy $\pi_{tot}^*$ into a combination of optimal local policies $\pi_i^*$.

**Proposition 4.3.** *Under the value decomposition scheme specified in Eq. (8) and assume the global behavior policy is decomposable (i.e., $\mu_{tot}(\boldsymbol{a}|\boldsymbol{o}) = \prod_{i=1}^{n} \mu_i(a_i|o_i)$), we have $\pi_{tot}^*(\boldsymbol{a}|\boldsymbol{o}) = \prod_{i=1}^{n} \pi_i^*(a_i|o_i)$, where $\pi_i^*$ is defined as:*

$$\pi_i^*(a_i|o_i) = \mu_i(a_i|o_i) \cdot \exp\left(\frac{w_i(\boldsymbol{o})}{\alpha}\left(Q_i^*(o_i, a_i) - V_i^*(o_i)\right)\right) \tag{9}$$

The result immediately follows by observing that:

$$\pi_{tot}^*(\boldsymbol{a}|\boldsymbol{o}) = \mu_{tot}(\boldsymbol{a}|\boldsymbol{o}) \cdot \exp\left(\frac{\sum_i w_i(\boldsymbol{o})\left(Q_i^*(o_i, a_i) - V_i^*(o_i)\right)}{\alpha}\right)$$

$$= \prod_{i=1}^{n} \mu_i(a_i|o_i) \cdot \exp\left(\frac{w_i(\boldsymbol{o})}{\alpha}\left(Q_i^*(o_i, a_i) - V_i^*(o_i)\right)\right) = \prod_{i=1}^{n} \pi_i^*(a_i|o_i)$$

$\pi_i^*$ can be perceived as the optimal local policy under the behavior regularization, since the RHS of Eq. (9) only depends on the optimal local value functions $Q_i^*$, $V_i^*$, and the local behavior policy $\mu_i$ of each agent $i$. This policy decomposition structure has a number of attractive characteristics. First, this design naturally bridges global value regularization and value decomposition. Second, $w_i(\boldsymbol{o})$ explicitly appears in the optimal local policy $\pi_i^*$, which is calculated from global observation $\boldsymbol{o}$. This makes the local policy optimizable with global information and consistent with the credit assignment in the multi-agent system.

### 4.3 Equivalent Implicit Local Value Regularizations

We have converted the relationship between global policies and values to the relationship between local policies and values. However, it is still unclear how to calculate the optimal local value functions in Eq. (9). Note that each local policy needs to satisfy $\sum_{a_i \in \mathcal{A}} \pi_i^*(a_i|o_i) = 1$ in Eq. (9) in order to ensure it is well-defined, it is also worth noting that when each local policy $\pi_i^*$ satisfies this self-normalization constraint, the optimal global policy $\pi_{tot}^* = \prod_i \pi_i^*$ will also satisfy $\sum_{\boldsymbol{a} \in \mathcal{A}^n} \pi_{tot}^*(\boldsymbol{a}|\boldsymbol{o}) = 1$. Thus, we only need to impose self-normalization constraints on local policies. Integrating the RHS of Eq. (9) over the local action space, and have:

$$\mathbb{E}_{a_i \sim \mu_i}\left[\exp\left(\frac{1}{\alpha}w_i(\boldsymbol{o})\left(Q_i^*(o_i, a_i) - V_i^*(o_i)\right)\right)\right] = 1 \tag{10}$$

We have now established the global-to-local relationships among the optimal values and policies, however, one annoying issue is that the global and local behavior policies $\mu_{tot}$ and $\mu_i$ are typically unknown, and it is hard to estimate them accurately, especially when they are multi-modal. However, perhaps surprisingly, we show that it is possible to learn optimal local value functions in an implicit way without knowing either $\mu_{tot}$ or $\mu_i$.

**Proposition 4.4.** *$V_i^*(\boldsymbol{o})$ can be obtained by solving the following convex optimization problem:*

$$\min_{V_i} \mathbb{E}_{a_i \sim \mu_i}\left[\exp\left(\frac{w_i(\boldsymbol{o})}{\alpha}\left(Q_i^*(o_i, a_i) - V_i(o_i)\right)\right) + \frac{w_i(\boldsymbol{o})V_i(o_i)}{\alpha}\right] \tag{11}$$

The proof follows that the first-order optimality condition of the above optimization objective (i.e., derivative with respect to $V_i$ equals 0) is exactly the condition Eq. (10).

The above optimization problem provides a new learning objective for the local state-value function $V_i$. It is also worth noting that this objective actually corresponds to adding an equivalent implicit local value regularization on $V_i$. To see this, note that in addition to performing expected regression to let $V_i$ fit $Q_i$ by the first term, this objective also minimizes the second term $w_i(\boldsymbol{o})V_i(o_i)/\alpha$ with respect to $V_i$ on behavioral data $\mu_i$. This is similar to performing conservative optimal value estimation in existing offline RL literature [20, 27] that learns an underestimated value function to avoid the distributional shift.

### 4.4 Algorithm Summary

Based on previous analysis, we are now ready to present our final algorithm, OMIGA. OMIGA consists of three supervised learning steps: learning local state-value function $V_i$, learning global and

---

**Algorithm 1** Pseudocode of OMIGA

---

**Require:** Offline dataset $\mathcal{D}$. hyperparameter $\alpha$.

1: Initialize local state-value network $V_i$, local action-value network $Q_i$ and its target network $\bar{Q}_i$, and policy network $\pi_i$ for agent $i$=1, 2, ... $n$.
2: Initialize the weight function network $w$ and $b$.
3: **for** $t = 1, \cdots$ , *max-value-iteration* **do**
4:      Sample batch transitions $(\boldsymbol{o}, \boldsymbol{a}, r, \boldsymbol{o}')$ from $\mathcal{D}$
5:      Update local state-value function $V_i(o_i)$ for each agent $i$ via Eq. (12).
6:      Compute $V_{tot}(\boldsymbol{o}')$, $Q_{tot}(\boldsymbol{o}, \boldsymbol{a})$ via Eq. (8).
7:      Update local action-value network $Q_i(o_i, a_i)$, weight function network $w(\boldsymbol{o})$ and $b(\boldsymbol{o})$ with objective Eq. (13).
8:      Update local policy network $\pi_i$ for each agent $i$ via Eq. (14).
9:      Soft update target network $\bar{Q}_i(o_i, a_i)$ by $Q_i(o_i, a_i)$ for each agent $i$.
10: **end for**

---

local Q-value functions, and learning local policies $\pi_i$. We can use Eq. (11) to learn the optimal local state-value function $V_i^*$ as:

$$\min_{V_i} \mathbb{E}_{(o_i, a_i) \sim \mathcal{D}} \left[ \exp \left( \frac{w_i(\boldsymbol{o})}{\alpha} \left( Q_i(o_i, a_i) - V_i(o_i) \right) \right) + \frac{w_i(\boldsymbol{o}) V_i(o_i)}{\alpha} \right] \tag{12}$$

With the learned local $V_i^*$, we can jointly optimize the local Q-value function $Q_i$, the weight and offset function $w_i$ and $b$ ($i = 1, \cdots, n$) with the following objective, where $Q_{tot}$ and $V_{tot}$ are computed based on the value decomposition in Eq. (8).

$$\min_{\substack{Q_i, w_i, b \\ i=1, \cdots, n}} \mathbb{E}_{(\boldsymbol{o}, \boldsymbol{a}, \boldsymbol{o}') \sim \mathcal{D}} \left[ \left( r(\boldsymbol{o}, \boldsymbol{a}) + \gamma V_{tot}\left(\boldsymbol{o}'\right) - Q_{tot}(\boldsymbol{o}, \boldsymbol{a}) \right)^2 \right] \tag{13}$$

After obtaining the optimal local value functions, we can further learn the local policies $\pi_i$ by minimizing the KL divergence between $\pi_i$ and $\pi_i^*$ in Eq. (9) [44], which yields the following learning objective:

$$\max_{\pi_i} \mathbb{E}_{(o_i, a_i) \sim \mathcal{D}} \left[ \exp \left( \frac{w_i(\boldsymbol{o})}{\alpha} \left( Q_i(o_i, a_i) - V_i(o_i) \right) \right) \cdot \log \pi_i(a_i | o_i) \right] \tag{14}$$

Through the above three learning steps, we convert the initial intractable global behavior regularization in multi-agent POMDP to tractable implicit value regularizations at the local level. Moreover, all three steps are learned in a completely in-sample manner, which performs supervised learning only on dataset samples without the involvement of potentially OOD policy-generated actions, thus greatly improving the training stability. We summarize the psuedocode of OMIGA in Algorithm 1.

**Discussion with prior works** OMIGA draws connections with several prior works such as ICQ [11], DMAC [45] and IVR [10]. ICQ bears some similarities with OMIGA, however, ICQ needs to estimate the normalizing partition function $Z(s) = \sum_a \mu(a|s) \exp\left(Q(s,a)/\alpha\right)$ to compute the weight in learning both $J_\pi$ and $J_Q$. $Z$ is estimated by learning an auxiliary behavior model or approximating with softmax operation over a mini-batch, which is hard to compute accurately, especially in the continuous action space. OMIGA uses Eq (11) to provide a new learning objective, it only needs to solve $V$ in a simple and elegant way by imposing self-normalization constraints. The difference between OMIGA and DMAC is that DMAC considers the KL divergence between current policy and previous policy, so that it can compute the KL divergence explicitly. While OMIGA considers the KL divergence between current policy and behavior policy, the behavior policy is typically unknown and hard to estimate, hence OMIGA proposes a principled way to implicitly compute the KL divergence. Compared with IVR which proposes a general implicit regularized RL framework in the offline single-agent setting, OMIGA starts from a similar regularized framework but is designed for the offline multi-agent setting. Also, OMIGA transforms the regularization from global to local in the multi-agent setting, this provides valuable insights to the community, justifies why previous offline multi-agent methods that apply local behavior regularization can work, and also reveals their limitations.

Table 1: Average scores and standard deviations over 5 random seeds on the offline multi-agent MuJoCo tasks and offline SMAC tasks.

| | | Multi-agent MuJoCo | | | | |
|---|---|---|---|---|---|---|
| Task | Dataset | BCQ-MA | CQL-MA | ICQ | OMAR | OMIGA(ours) |
| Hopper | expert | 77.85±58.04 | 159.14± 313.83 | 754.74± 806.28 | 2.36± 1.46 | **859.63±709.47** |
| Hopper | medium | 44.58±20.62 | 401.27±199.88 | 501.79±14.03 | 21.34±24.90 | **1189.26± 544.30** |
| Hopper | medium-replay | 26.53±24.04 | 31.37±15.16 | 195.39±103.61 | 3.30±3.22 | **774.18±494.27** |
| Hopper | medium-expert | 54.31±23.66 | 64.82±123.31 | 355.44±373.86 | 1.44±0.86 | **709.00±595.66** |
| Ant | expert | 1317.73±286.28 | 1042.39±2021.65 | 2050.00±11.86 | 312.54±297.48 | **2055.46±1.58** |
| Ant | medium | 1059.60±91.22 | 533.90±1766.42 | 1412.41±10.93 | -1710.04±1588.98 | **1418.44±5.36** |
| Ant | medium-replay | 950.77±48.76 | 234.62±1618.28 | 1016.68±53.51 | -2014.20±844.68 | **1105.13±88.87** |
| Ant | medium-expert | 1020.89±242.74 | 800.22±1621.52 | 1590.18±85.61 | -2992.80± 6.95 | **1720.33±110.63** |
| HalfCheetah | expert | 2992.71±629.65 | 1189.54±1034.49 | 2955.94±459.19 | -206.73±161.12 | **3383.61±552.67** |
| HalfCheetah | medium | 2590.47±1110.35 | 1011.35±1016.94 | 2549.27±96.34 | -265.68±146.98 | **3608.13±237.37** |
| HalfCheetah | medium-replay | -333.64±152.06 | 1998.67±693.92 | 1922.42±612.87 | -235.42±154.89 | **2504.70±83.47** |
| HalfCheetah | medium-expert | **3543.70±780.89** | 1194.23±1081.04 | 2833.99±420.32 | -253.84± 63.94 | 2948.46± 518.89 |
| | | SMAC | | | | |
| Task | Dataset | BCQ-MA | CQL-MA | ICQ | OMAR | OMIGA(ours) |
| 5m_vs_6m | good | 7.76±0.15 | 8.08±0.21 | 7.87±0.30 | 7.40±0.63 | **8.25±0.37** |
| 5m_vs_6m | medium | 7.58±0.10 | 7.78±0.10 | 7.77±0.3 | 7.08±0.51 | **7.92±0.57** |
| 5m_vs_6m | poor | **7.61±0.36** | 7.43±0.10 | 7.26±0.19 | 7.27±0.42 | 7.52±0.21 |
| 2c_vs_64zg | good | 19.13±0.27 | 18.48±0.95 | 18.82±0.17 | 17.27±0.78 | **19.15±0.32** |
| 2c_vs_64zg | medium | 15.58±0.37 | 12.82±1.61 | 15.57±0.61 | 10.20±0.20 | **16.03±0.19** |
| 2c_vs_64zg | poor | 12.46±0.18 | 10.83±0.51 | 12.56±0.18 | 11.33±0.50 | **13.02±0.66** |
| 6h_vs_8z | good | 12.19±0.23 | 10.44±0.20 | 11.81±0.12 | 9.85±0.28 | **12.54±0.21** |
| 6h_vs_8z | medium | 11.77±0.16 | 11.29±0.29 | 11.13±0.33 | 10.36±0.16 | **12.19±0.22** |
| 6h_vs_8z | poor | 10.84±0.16 | 10.81±0.52 | 10.55±0.10 | 10.63±0.25 | **11.31±0.19** |
| corridor | good | 15.24±1.21 | 5.22±0.81 | 15.54±1.12 | 6.74±0.69 | **15.88±0.89** |
| corridor | medium | 10.82±0.92 | 7.04±0.66 | 11.30±1.57 | 7.26±0.71 | **11.66±1.30** |
| corridor | poor | 4.47±0.94 | 4.08±0.60 | 4.47±0.33 | 4.28±0.49 | **5.61±0.35** |

## 5 Experiment

### 5.1 Offline Multi-Agent Datasets

We choose multi-agent MuJoCo [15] and StarCraft Multi-Agent Challenge (SMAC) [16] as our experiment environments. Multi-agent MuJoCo is a benchmark for continuous multi-agent robotic control, based on MuJoCo environment. Currently, there is still a lack of offline MARL datasets with continuous action. Thus, we provide an offline multi-agent MuJoCo dataset which is collected from HAPPO [46] algorithm. The dataset contains multiple different data types and the details of the dataset are shown in Appendix B. Another benchmark used in this paper is SMAC, a popular multi-agent cooperative environment for evaluating advanced MARL methods. The offline dataset we used is provided by Meng et al. [38], which is collected from the online trained MAPPO agents [47], and is the largest open offline dataset on SMAC. The offline dataset has three quality levels: good, medium, and poor. Besides, SMAC includes several StarCraft II multi-agent micromanagement maps. We consider 4 representative battle maps: 2 hard maps (5m_vs_6m, 2c_vs_64zg), and 2 super hard maps (6h_vs_8z, corridor).

### 5.2 Baselines

We compare OMIGA with four recent offline MARL algorithms: the multi-agent version of BCQ [4] and CQL [20] (namely BCQ-MA and CQL-MA), ICQ [11] and OMAR [12]. BCQ-MA and CQL-MA use linear weighted value decomposition structure as Eq.(1) for the multi-agent setting. Details for the implementations of baseline algorithms and the hyperparameters used in OMIGA and baselines are shown in Appendix C.

### 5.3 Comparative Evaluation

Table 1 shows the mean and standard deviation of average returns for the offline Multi-agent MuJoCo and SMAC tasks. Each algorithm is evaluated by using 32 independent episodes and runs with 5 seeds for training. The results show that OMIGA outperforms all baselines and achieves state-of-the-art performance in most tasks.

For these multi-agent tasks, the credit assignment for multi-agent is complex and it is difficult to learn multi-agent cooperation in the offline setting. The global-to-local value regularization design of

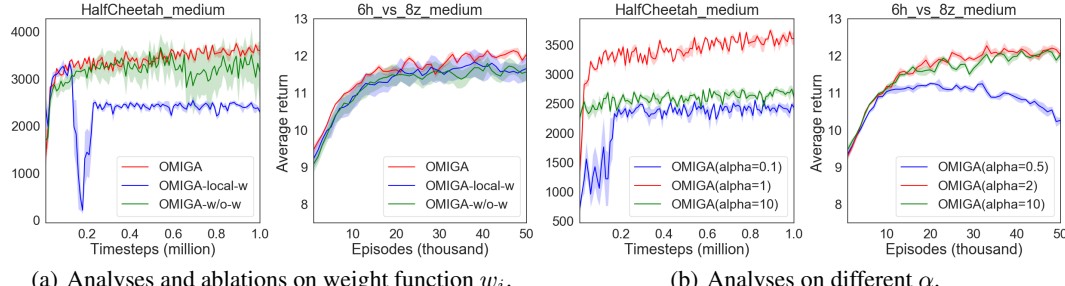

(a) Analyses and ablations on weight function $w_i$.      (b) Analyses on different $\alpha$.

Figure 1: Analyses and ablations.

our method reflects the global regularization in the local value function learning implicitly, which leads to better performance than other baseline algorithms. Moreover, the state-value function and Q-value function in OMIGA are completely learned in an in-sample manner without the involvement of the agent policy, which also leads to better training stability and performance. Besides, compared with baseline algorithms, the local policy learning in OMIGA contains global information, and thus enjoys obvious advantages in tasks with continuous action space like multi-agent MuJoCo.

We also conduct experiments on more diverse datasets which are obtained by mixing datasets of various quality. The results are shown in Appendix D. On these mixed datasets, the behavior policies can be suboptimal and more heterogeneous. Therefore, it is difficult for algorithms such as BCQ-MA and OMAR to learn an accurate behavior policy, making implicit value learning with the regularization framework of OMIGA more appealing.

## 5.4   Analyses on Policy Learning with Global Information

As Eq. (12) and Eq. (14) show, the learning of both local state-value function $V_i$ and local policy $\pi_i$ in OMIGA contains global information, which is reflected in $w_i(\boldsymbol{o})$. Therefore, OMIGA can make the credit assignment during value and policy learning and leverage the global information to improve local policy. To verify the effect of this design, we transform OMIGA into two versions without global information, and then compare their performance on offline multi-agent MuJoCo and SMAC datasets. In OMIGA-w/o-w, local policy learning uses formula $\max_{\pi_i} \mathbb{E}_{(o_i,a_i)\sim\mathcal{D}}[\exp(\frac{1}{\alpha}(Q_i(o_i,a_i)-V_i(o_i)))\cdot \log \pi_i(a_i|o_i)]$ and the weight function is not available during the policy learning. In OMIGA-local-w, the weight function is $w_i(o_i)$ that only contains local information, and local policy learning uses formula $\max_{\pi_i} \mathbb{E}_{(o_i,a_i)\sim\mathcal{D}}[\exp(\frac{w_i(o_i)}{\alpha}(Q_i(o_i,a_i)-V_i(o_i)))\cdot \log \pi_i(a_i|o_i)]$.

Figure 1(a) shows the experimental results on offline HalfCheetah datasets and 6h_vs_8z datasets. For these difficult multi-agent tasks, the cooperative relationships among agents are extremely complex, so global information is important to guide local policy learning. OMIGA-w/o-w and OMIGA-local-w lack global information, and policies are only learned at the local level, which causes worse performance and stability.

## 5.5   Analyses on the Regularization Hyperparameter

In OMIGA, hyperparameter $\alpha$ is used to control the degree of regularization. The higher $\alpha$ encourages the policy to stay closer to the behavioral distribution, and the lower $\alpha$ makes the policy more aggressive and optimistic. Figure 1(b) shows the experiments about $\alpha$ on the offline MuJoCo HalfCheetah task and SMAC 6h_vs_8z task. In the HalfCheetah task, too high and too low $\alpha$ will lead to performance degradation. An appropriate $\alpha$ can ensure that the policy is restricted near the dataset distribution and also does not lose performance due to being too conservative. In the 6h_vs_8z task, the dataset contains enough good transitions. Therefore, the degree of regularization should be higher to make the algorithm more conservative. Experimental results indicate that a higher $\alpha$ brings better performance on such datasets. When $\alpha$ is very small, the performance of the algorithm becomes very poor due to the lack of value regularization.

## 6   Conclusion

In this paper, we study the key challenge of the offline MARL problem. We start from the usage of global behavior constraints and smartly derive a global-to-local value regularization scheme under value decomposition. By doing so, we naturally get a new offline MARL algorithm, OMIGA, that in principle applies global-level regularization but actually imposes equivalent implicit local-level regularization. Our work reveals why the local-level regularization used by existing algorithms works and gives theoretical insights into their weaknesses and how to develop better algorithms. The global-to-local regularization design of OMIGA can capture global information and the impact of multi-agent credit assignment under the offline setting, which guarantees that the learned local policies are jointly optimal at the global level. One future work is to develop other offline MARL algorithms by using other choices of $f$ functions.

## Acknowledgments

This work is supported by National Key Research and Development Program of China under Grant (2022YFB2502904) and funding from Global Data Solutions Limited.

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

# A    Proofs

## A.1    Proof of Theorem 4.1

*Proof.* For any two global state-value functions $V_{tot}^1$ and $V_{tot}^2$, let $\pi_{tot}^1$ be the optimal global policy under $\mathcal{T}_f^* V_{tot}^1$, and we can get:

$$
\begin{aligned}
&\left(\mathcal{T}_f^* V_{tot}^1\right)(\boldsymbol{o}) - \left(\mathcal{T}_f^* V_{tot}^2\right)(\boldsymbol{o}) \\
&= \sum_{\boldsymbol{a}} \pi_{tot}^1(\boldsymbol{a}|\boldsymbol{o}) \left[ r + \gamma \mathbb{E}_{\boldsymbol{o}'} \left[ V_{tot}^1 \left(\boldsymbol{o}'\right) \right] - \alpha \log \left( \frac{\pi_{tot}^1(\boldsymbol{a}|\boldsymbol{o})}{\mu_{tot}(\boldsymbol{a}|\boldsymbol{o})} \right) \right] - \max_{\pi_{tot}} \sum_{\boldsymbol{a}} \pi_{tot}(\boldsymbol{a}|\boldsymbol{o}) \left[ r + \gamma \mathbb{E}_{\boldsymbol{o}'} \left[ V_{tot}^2 \left(\boldsymbol{o}'\right) \right] - \alpha \log \left( \frac{\pi_{tot}(\boldsymbol{a}|\boldsymbol{o})}{\mu_{tot}(\boldsymbol{a}|\boldsymbol{o})} \right) \right] \\
&\leq \sum_{\boldsymbol{a}} \pi_{tot}^1(\boldsymbol{a}|\boldsymbol{o}) \left[ r + \gamma \mathbb{E}_{\boldsymbol{o}'} \left[ V_{tot}^1 \left(\boldsymbol{o}'\right) \right] - \alpha \log \left( \frac{\pi_{tot}^1(\boldsymbol{a}|\boldsymbol{o})}{\mu_{tot}(\boldsymbol{a}|\boldsymbol{o})} \right) \right] - \sum_{\boldsymbol{a}} \pi_{tot}^1(\boldsymbol{a}|\boldsymbol{o}) \left[ r + \gamma \mathbb{E}_{\boldsymbol{o}'} \left[ V_{tot}^2 \left(\boldsymbol{o}'\right) \right] - \alpha \log \left( \frac{\pi_{tot}^1(\boldsymbol{a}|\boldsymbol{o})}{\mu_{tot}(\boldsymbol{a}|\boldsymbol{o})} \right) \right] \\
&= \gamma \sum \pi_{tot}^1(\boldsymbol{a}|\boldsymbol{o}) \mathbb{E}_{\boldsymbol{o}'} \left[ V_{tot}^1 \left(\boldsymbol{o}'\right) - V_{tot}^2 \left(\boldsymbol{o}'\right) \right] \\
&\leq \gamma \left\| V_{tot}^1 - V_{tot}^2 \right\|_{\infty}
\end{aligned}
$$

Therefore, it follows that:

$$
\left\| \mathcal{T}_f^* V_{tot}^1 - \mathcal{T}_f^* V_{tot}^2 \right\|_{\infty} \leq \gamma \left\| V_{tot}^1 - V_{tot}^2 \right\|_{\infty}
$$

$\square$

## A.2    Proof of Proposition 4.2

*Proof.* For a behavior-regularized multi-agent POMDP with the behavior regularizer function $f(\pi_{tot}, \mu_{tot}) = \log(\pi_{tot}/\mu_{tot})$, its learning objective can be expressed as $\max_{\pi_{tot}} \mathbb{E} \left[ \sum_{t=0}^{\infty} \gamma^t \left( r\left(\boldsymbol{o}_t, \boldsymbol{a}_t\right) - \alpha log\left(\pi_{tot}\left(\boldsymbol{a}_t|\boldsymbol{o}_t\right) / \mu_{tot}\left(\boldsymbol{a}_t|\boldsymbol{o}_t\right)\right)\right) \right]$. Its Lagrangian function can obtain when the optimal global policy is written as follows:

$$
\begin{aligned}
L(\pi_{tot}, \beta, u) = &\sum_{\boldsymbol{o}} d_{\pi_{tot}}(\boldsymbol{o}) \sum_{\boldsymbol{a}} \pi_{tot}(\boldsymbol{a}|\boldsymbol{o}) \left( Q_{tot}(\boldsymbol{o}, \boldsymbol{a}) - \alpha \log \left( \frac{\pi_{tot}(\boldsymbol{a}|\boldsymbol{o})}{\mu_{tot}(\boldsymbol{a}|\boldsymbol{o})} \right) \right) \\
&- \sum_{\boldsymbol{o}} d_{\pi_{tot}}(\boldsymbol{o}) \left[ u(\boldsymbol{o}) \left( \sum_{\boldsymbol{a}} \pi_{tot}(\boldsymbol{a}|\boldsymbol{o}) - 1 \right) + \sum_{\boldsymbol{a}} \beta(\boldsymbol{a}|\boldsymbol{o}) \pi_{tot}(\boldsymbol{a}|\boldsymbol{o}) \right],
\end{aligned}
$$

where $d_{\pi_{tot}}$ is the stationary joint observation distribution of the global policy $\pi_{tot}$. $u$ and $\beta$ are Lagrangian multipliers for the equality and inequality constraints.

According to the Karush-Kuhn-Tucker (KKT) conditions where the derivative of the Lagrangian objective function with respect to the global policy is zero at the optimal solution, it follows that:

$$
Q_{tot}(\boldsymbol{o}, \boldsymbol{a}) - \alpha \left( \log \left( \frac{\pi_{tot}(\boldsymbol{a}|\boldsymbol{o})}{\mu_{tot}(\boldsymbol{a}|\boldsymbol{o})} \right) + 1 \right) - u(\boldsymbol{o}) + \beta(\boldsymbol{a}|\boldsymbol{o}) = 0 \tag{15}
$$

$$
\sum_{\boldsymbol{a}} \pi_{tot}(\boldsymbol{a}|\boldsymbol{o}) = 1
$$

$$
\beta(\boldsymbol{a}|\boldsymbol{o}) \pi_{tot}(\boldsymbol{a}|\boldsymbol{o}) = 0
$$

$$
0 \leq \pi_{tot}(\boldsymbol{a}|\boldsymbol{o}) \leq 1 \text{ and } 0 \leq \beta(\boldsymbol{a}|\boldsymbol{o})
$$

From Eq. (15), we can further solve the optimal global policy as:

$$
\pi_{tot}(\boldsymbol{a}|\boldsymbol{o}) = \mu_{tot}(\boldsymbol{a}|\boldsymbol{o}) \cdot \exp \left( \frac{Q_{tot}(\boldsymbol{o}, \boldsymbol{a}) - u(\boldsymbol{o}) + \beta(\boldsymbol{a}|\boldsymbol{o})}{\alpha} - 1 \right)
$$

The above formula can be further simplified. $\beta$ is the Lagrangian multiplier, and meets complementary slackness $\beta(\boldsymbol{a}|\boldsymbol{o}) \pi_{tot}(\boldsymbol{a}|\boldsymbol{o}) = 0$. Considering the joint observation $\boldsymbol{o}$ is fixed, $\exp \left( \frac{Q_{tot}(\boldsymbol{o}, \boldsymbol{a}) - u(\boldsymbol{o}) + \beta(\boldsymbol{a}|\boldsymbol{o})}{\alpha} - 1 \right)$ is always larger than 0. Therefore, for any positive probability action, its corresponding Lagrangian multiplier $\beta(\boldsymbol{a}|\boldsymbol{o})$ is 0. Therefore, $\pi_{tot}(\boldsymbol{a}|\boldsymbol{o})$ can be reformulated as:

$$
\pi_{tot}(\boldsymbol{a}|\boldsymbol{o}) = \mu_{tot}(\boldsymbol{a}|\boldsymbol{o}) \cdot \exp \left( \frac{Q_{tot}(\boldsymbol{o}, \boldsymbol{a}) - u(\boldsymbol{o})}{\alpha} - 1 \right) \tag{16}
$$

Bringing Eq. (16) into $\sum_{\boldsymbol{a}} \pi_{tot}(\boldsymbol{a}|\boldsymbol{o}) = 1$, we have:

$$\mathbb{E}_{\boldsymbol{a} \sim \mu_{tot}} \left[ \exp \left( \frac{Q_{tot}(\boldsymbol{o}, \boldsymbol{a}) - u(\boldsymbol{o})}{\alpha} - 1 \right) \right] = 1 \qquad (17)$$

The left side of Eq. (17) can be seen as a continuous and monotonic function of $u$, so it has only one solution denoted as $u^*$, and we denote the corresponding policy $\pi_{tot}$ as $\pi_{tot}^*$.

Integrating Eq. (16) into the expression of optimal global state value, we can get:

$$
\begin{aligned}
V_{tot}^*(\boldsymbol{o}) &= \mathcal{T}_f^* V_{tot}^*(\boldsymbol{o}) \\
&= \sum_{\boldsymbol{a}} \pi_{tot}^*(\boldsymbol{a}|\boldsymbol{o}) \left( Q_{tot}^*(\boldsymbol{o}, \boldsymbol{a}) - \alpha \log \left( \frac{\pi_{tot}^*(\boldsymbol{a}|\boldsymbol{o})}{\mu_{tot}(\boldsymbol{a}|\boldsymbol{o})} \right) \right) \\
&= \sum_{\boldsymbol{a}} \pi_{tot}^*(\boldsymbol{a}|\boldsymbol{o}) \left( u^*(\boldsymbol{o}) + \alpha \right) \\
&= u^*(\boldsymbol{o}) + \alpha
\end{aligned}
$$

To summarize, we obtain the optimality condition of the behavior regularized MDP with Reverse KL divergence as follows:

$$
\begin{aligned}
Q_{tot}^*(\boldsymbol{o}, \boldsymbol{a}) &= r(\boldsymbol{o}, \boldsymbol{a}) + \gamma \mathbb{E}_{\boldsymbol{o}'|\boldsymbol{o}, \boldsymbol{a}} \left[ V_{tot}^*(\boldsymbol{o}') \right] \\
V_{tot}^*(\boldsymbol{o}) &= u^*(\boldsymbol{o}) + \alpha \\
\pi_{tot}^*(\boldsymbol{a}|\boldsymbol{o}) &= \mu_{tot}(\boldsymbol{a}|\boldsymbol{o}) \cdot \exp \left( \frac{Q_{tot}^*(\boldsymbol{o}, \boldsymbol{a}) - u^*(\boldsymbol{o})}{\alpha} - 1 \right)
\end{aligned}
$$

where $u(\boldsymbol{o})$ is a normalization term and has a optimal value $u^*$ that makes the corresponding optimal policy $\pi_{tot}^*$ satisfy $\sum_{\boldsymbol{a} \in \mathcal{A}^n} \pi_{tot}^*(\boldsymbol{a}|\boldsymbol{o}) = 1$.

$\square$

# B  Experiment Settings

## B.1  Multi-Agent MuJoCo

Multi-agent Mujoco [15] is a benchmark framework developed for assessing and comparing the effectiveness of algorithms in continuous multi-agent robotic control. Within this framework, a robotic system is partitioned into independent agents, each tasked with controlling a specific set of joints. The agents collaborate harmoniously to accomplish shared objectives, such as acquiring the ability to walk through an environment, with the ultimate goal of maximizing the cumulative reward. Multi-agent MuJoCo environment consists of multiple different robot configurations, and it is often used for the study of novel MARL algorithms for decentralized coordination in isolation.

To generate the dataset transitions, we captured the interactions between the environment and trained online MARL algorithms. Specifically, we use HAPPO [46] algorithm to collect data. The expert dataset is generated by employing the converged HAPPO algorithm. This involves training the algorithm until it reaches a state of convergence, where the agents have learned optimal policies. The medium dataset is generated by first training a policy online using HAPPO, early-stopping the training, and collecting samples from this partially-trained policy. The medium-replay dataset consists of recording all samples in the replay buffer observed during training until the policy reaches the medium level of performance. The medium-expert dataset is constructed by mixing equal amounts of expert demonstrations and suboptimal data. For all datasets, the hyperparameter env_args.agent_obsk (determines up to which connection distance agents will be able to form observations) is set to 1. The average returns of our datasets are listed in Table 2.

## B.2  The StarCraft Multi-Agent Challenge

The StarCraft Multi-Agent Challenge (SMAC) benchmark is chosen as our testing environment. Due to its high control complexity, SMAC is a popular multi-agent cooperative control environment for evaluating advanced MARL methods. It consists of a collection of StarCraft II micro-management

Table 2: The multi-agent MuJoCo datasets.

| Scenario | Quality | Average Return |
|----------|---------|----------------|
| 2-Agent Ant | expert | 2055.07 |
| | medium | 1418.70 |
| | medium-expert | 1736.88 |
| | medium-replay | 1029.51 |
| 3-Agent Hopper | expert | 2452.02 |
| | medium | 723.57 |
| | medium-expert | 1190.61 |
| | medium-replay | 746.42 |
| 6-Agent HalfCheetah | expert | 2785.10 |
| | medium | 1425.66 |
| | medium-expert | 2105.38 |
| | medium-replay | 655.76 |

tasks in which two groups of units engage in combat. Agents based on the MARL algorithm control the first group's units, while a built-in heuristic game AI bot with different difficulties controls the second group's units. Scenarios vary in terms of the initial location, number and type of units, and elevated or impassable terrain. The available actions for each agent include no operation, move[direction], attack [enemy id], and stop. The reward that each agent receives is the same. The hit-point damage dealt and received determines the agents' share of the reward. SMAC consists of several StarCraft II multi-agent micromanagement maps. We consider 4 representative battle maps, including 2 hard maps (5m_vs_6m, 2c_vs_64zg), and 2 super hard maps (6h_vs_8z, corridor), as our experiment tasks. The task type and other details of the maps are listed in Table 3.

Table 3: SMAC maps for experiments.

| Map Name | Ally Units | Enemy Units | Type |
|----------|-----------|-------------|------|
| 5m_vs_6m | 5 Marines | 6 Marines | homogeneous & asymmetric |
| 2c_vs_64zg | 2 Colossi | 64 Zerglings | micro-trick: positioning |
| 6h_vs_8z | 6 Hydralisks | 8 Zealots | micro-trick: focus fire |
| corridor | 6 Zealots | 24 Zerglings | micro-trick: wall off |

The offline SMAC dataset used in this study is provided by [38], which is the largest open offline dataset on SMAC. Different from single-agent offline datasets, it considers the property of multi-agent POMDP, which owns local observations and available actions for each agent. The dataset is collected from the trained MAPPO agent and includes three quality levels: good, medium, and poor. For each original large dataset, we randomly sample 1000 episodes as our dataset. The average returns of SMAC datasets are listed in Table 4.

Table 4: The SMAC datasets.

| Map Name | Quality | Average Return |
|----------|---------|----------------|
| 5m_vs_6m | good | 20.00 |
| | medium | 11.03 |
| | poor | 8.50 |
| 2c_vs_64zg | good | 19.94 |
| | medium | 13.00 |
| | poor | 8.89 |
| 6h_vs_8z | good | 17.84 |
| | medium | 11.96 |
| | poor | 9.12 |
| corridor | good | 19.88 |
| | medium | 13.07 |
| | poor | 4.93 |

# C   Implementation Details

## C.1   Details of OMIGA

The local Q-value, state-value networks and policy networks of OMIGA are represented by 3-layer ReLU activated MLPs with 256 units for each hidden layer. For the weight network, we use 2-layer ReLU-activated MLPs with 64 units for each hidden layer. All the networks are optimized by Adam optimizer.

## C.2   Details of baselines

We compare OMIGA against four recent offline MARL algorithms: ICQ [11], OMAR [12], BCQ-MA, and CQL-MA. For ICQ and OMAR, we implement them based on the algorithm described in their papers. BCQ-MA is the multi-agent version of BCQ [4], and CQL-MA is the multi-agent version of CQL [20]. BCQ-MA and CQL-MA use the linear weighted value decomposition structure as $Q_{tot} = \sum_{i=1}^{n} w_i(\boldsymbol{o})Q_i(o_i, a_i) + b(\boldsymbol{o}), w^i \geq 0$ for the multi-agent setting. The policy constraint of BCQ-MA and the value regularization of CQL-MA are both imposed on the local Q-value.

In this paper, all experiments are implemented with Pytorch and executed on NVIDIA V100 GPUs.

## C.3   Hyperparameters

For multi-agent MuJoCo, the hyperparameters of OMIGA and baselines are listed in Table 5. An important hyperparameter of OMIGA is the regularization hyperparameter $\alpha$. The higher $\alpha$ encourages OMIGA to stay near the behavioral distribution, and lower $\alpha$ makes OMIGA more optimistic. On most tasks, we use $\alpha = 10$ to ensure a good regularization effect. On the medium-quality dataset of the HalfCheetah task, we choose $\alpha = 1$.

Table 5: Hyperparameters of OMIGA and baselines for multi-agent MuJoCo.

| Hyperparameter | Value |
|---|---|
| **Shared parameters** | |
| Q-value network learning rate | 5e-4 |
| Policy network learning rate | 5e-4 |
| Optimizer | Adam |
| Target update rate | 0.005 |
| Batch size | 128 |
| Discount factor | 0.99 |
| Hidden dimension | 256 |
| Weight network hidden dimension | 64 |
| **OMIGA** | |
| State-value network learning rate | 5e-4 |
| Regularization parameter $\alpha$ | 1 or 10 |
| **Others** | |
| Lagrangian coefficient (ICQ) | 10 |
| Tradeoff factor $\alpha$ (OMAR, CQL-MA) | 1 |

For SMAC, the hyperparameters of OMIGA and baselines are listed in Table 6. On most tasks, we use $\alpha = 10$. On the poor dataset of the 6h_vs_8z map, the quality of the dataset is relatively poor. It does not make much sense to make the policy close to the behavioral policy, so we choose $\alpha = 2$ to make the algorithm more radical.

On most SMAC maps, the learning rate of all networks is set to 5e-4. The exception is the map 2c_vs_64zg, on this map, the learning rate of all networks is set to 1e-4.

Table 6: Hyperparameters of OMIGA and baselines for SMAC.

| Hyperparameter | Value |
|---|---|
| **Shared parameters** | |
| Q-value network learning rate | 5e-4 or 1e-4 |
| Policy network learning rate | 5e-4 or 1e-4 |
| Optimizer | Adam |
| Target update rate | 0.005 |
| Batch size | 128 |
| Discount factor | 0.99 |
| Hidden dimension | 256 |
| Weight network hidden dimension | 64 |
| **OMIGA** | |
| State-value network learning rate | 5e-4 or 1e-4 |
| Regularization parameter $\alpha$ | 2 or 10 |
| **Others** | |
| Lagrangian coefficient (ICQ) | 10 |
| Threshold (BCQ-MA) | 0.3 |
| Tradeoff factor $\alpha$ (OMAR, CQL-MA) | 1 |

Table 7: Average scores and standard deviations over 5 random seeds on the mixed offline SMAC datasets.

| Map | Dataset | BCQ-MA | CQL-MA | ICQ | OMAR | OMIGA(ours) |
|---|---|---|---|---|---|---|
| 6h_vs_8z | good-poor | 11.41±0.44 | 9.56±0.25 | 11.00±0.36 | 9.17±0.19 | **11.88±0.27** |
| 6h_vs_8z | good-medium | 11.79±0.29 | 10.08±0.26 | 11.18±0.25 | 10.02±0.16 | **12.05±0.47** |
| 6h_vs_8z | medium-poor | 11.18±0.41 | 10.73±0.38 | 11.25±0.35 | 10.42±0.19 | **11.85±0.35** |
| corridor | good-poor | 12.37±1.36 | 4.88±0.35 | 11.78±1.53 | 5.54±0.75 | **13.01±0.89** |
| corridor | good-medium | 13.32±0.71 | 5.77±1.30 | 12.98±0.62 | 6.63±0.74 | **14.02±1.04** |
| corridor | medium-poor | 8.11±0.35 | 6.18±0.59 | 8.27±0.48 | 6.25 ±0.48 | **9.70±1.40** |

# D   Additional Results

## D.1   Results on mixed datasets

We also evaluated the performance of OMIGA on the multi-modal mixed datasets. Unlike BCQ-MA and OMAR, OMIGA doesn't need to learn a behavior policy. We choose two original datasets on the SMAC super hard maps 6h_vs_8z and corridor and construct mixed datasets by combining these SMAC datasets of different quality, including good-poor, good-medium, and medium-poor datasets. Each mixed dataset is blended by 50% of each of the two original datasets. On these mixed suboptimal datasets, the behavior policy is heterogeneous. Therefore, it is more difficult for algorithms such as BCQ-MA and OMAR to learn an accurate behavior policy, making implicit value learning with the regularization framework of OMIGA more appealing.

Table 7 shows the results that OMIGA consistently outperforms other offline MARL baselines under all different mixed dataset experiments. Compared with the results on the original datasets, the performance of OMIGA has become more leading, indicating the benefits of implicit value regularization of OMIGA.

