# OpenReview forum: "Offline Multi-Agent Reinforcement Learning with Implicit Global-to-Local Value Regularization"
_NeurIPS.cc/2023/Conference — NeurIPS 2023 poster_

### Official Review · Reviewer_uuTt · 2023-06-15

**Soundness:** 3 good
**Presentation:** 3 good
**Contribution:** 2 fair
**Rating:** 5
**Confidence:** 2

**Summary:**

The authors provide a principled framework to convert global-level value regularization into equivalent implicit local value regularizations and simultaneously enable in-sample learning, thus elegantly bridging multi-agent value decomposition and policy learning with offline regularizations. The method is evaluated through both theoretical analysis and experiments conducted on the offline multi-agent MuJoCo and StarCraft II micro-management tasks.


**Strengths:**

1. The methods are evaluated through both theoretical analysis and experiments.
2. The writing is clear and easily comprehensible.

**Weaknesses:**

1. The paper's main contribution is to decompose the policy and value function based on the factorization of the behavior policies. However, the factorization assumption is not well justified. The author should provide more analysis and some simple examples to illustrate the assumption.
2. Could the authors analyze whether the local Q-function Q_i adheres to the Individual-Global-Maximum (IGM)?
3. Do all of the compared methods have access to global information? If not, the fairness of the results may be compromised.
4. Another point worth considering is that in homogeneous scenarios, where agents exhibit consistent behavior, would there still be issues with simply enforcing local-level regularization?

**Questions:**

Please answer the questions in the weaknesses.

---

> ### Author Rebuttal · Authors · 2023-08-09
>
> Thank you for your review.
>
> 1.**The factorization of the behavior policies.**  In multi-agent reinforcement learning,  the global policy is usually considered decomposable,  which means $\pi_{tot}(\boldsymbol{a} |\boldsymbol{o})=\prod_{i=1}^n \pi_i\left(a_i | o_i\right)$, see [1] [2] [3] [4]. The global  behavior policy also belongs to a type of global policy, so it is reasonable to believe that the global behavior policy can be decomposed.
>
> 2.**Whether the local Q-function Q_i adheres to the Individual-Global-Maximum.** Our decomposition of Q also satisfies $Q_{t o t}(\boldsymbol{o}, \boldsymbol{a})=\sum_i w_i(\boldsymbol{o}) Q_i\left(o_i, a_i\right)+b(\boldsymbol{o}), w_i \geq 0, \forall i=1 \cdots, n$ . Due to the nonnegativity of the weight function $w_i$ ,  when each local Q reaches the action that maximizes the local Q-value,  the global Q obtains the maximum joint action.
>
> 3.**Do all of the compared methods have access to global information?** Yes, all the baseline methods have access to the global information. The baseline methods  adopt the same Q decomposition structure as OMIGA, and the global information is reflected in the weight $w_i(\boldsymbol{o})$ , which is calculated based on the joint observation.
>
> 4.**Simply enforcing local-level regularization in homogeneous scenarios.**  Even in homogeneous scenarios, the multi-agent system remains a whole. Thus,  global-level regularization is still better than local-level regularization, because it is difficult for the local-level offline regularization to consider the global information that captures coordinated agent behaviors and the credit assignment in the multi-agent system. Besides, for most homogeneous scenarios,  even if the action space of agents is the same, the behavior of agents is still diverse. Simply enforcing local-level regularization cannot guarantee the regularization still remains valid at the global level, and it also potentially leads to over-conservative policy learning.
>
>
> [1] Divergence-Regularized Multi-Agent Actor-Critic,  ICML 2022
>
> [2] Revisiting Some Common Practices in Cooperative Multi-Agent Reinforcement Learning, ICML 2022
>
> [3] More Centralized Training, Still Decentralized Execution: Multi-Agent Conditional Policy Factorization, ICLR 2023
>
> [4] Dealing with Non-Stationarity in MARL via Trust-Region Decomposition, ICLR 2022

---

> > ### Comment · Reviewer_uuTt · 2023-08-17
> >
> > Thank you for the response.

---

### Official Review · Reviewer_WVBc · 2023-06-15

**Soundness:** 3 good
**Presentation:** 3 good
**Contribution:** 3 good
**Rating:** 5
**Confidence:** 5

**Summary:**

The paper investigates multi-agent offline reinforcement learning, and address some of the problems that arise with applying regularization at either the global or local level. To achieve this, a framework that allows to apply global regularization via an equivalent set of local regularizations is proposed. This way, the CTDE paradigm and value-decomposition scheme can be applied, and decentralized policy are easily learned, while at the same time retaining the positive effect and guarantees that global regularization has.

**Strengths:**

The paper investigates an interesting area: offline reinforcement learning can be a viable way to learn in those real-world settings where actual environment interaction is not possible or too expensive. The application of the CTDE paradigm to effectively translate the global terms into local terms that are easier to learn and use is well aligned with the current research track on this line. The resulting algorithm is easy to follow and implement, and its performances are shown to be superior to other exiting offline RL algorithms with more complex regularization terms. Also, the paper is nicely written and clear to follow, and the empirical results are quite convincing.

**Weaknesses:**

However, the paper in my opinion is a bit sloppy when it comes to a proper characterization of the used setting, and the actual working mechanism of the various components (policies, value-functions...) that are used. A precise mathematical formulation is a very useful point for a paper, as it allows the reader to relate the present work to the previous literature. Please see my questions below for a more precise explanation.

**Questions:**

- In the usual Dec-POMDP framework, the global Q-function does not condition on the joint observation $\mathbf{o}$, but rather on the joint action-observation history $\tau$ (and/or on the true environment state $s$). This is because the joint observation $\mathbf{o}$ is not a sufficient statistics for the agents to learn an optimal policy $\pi_{tot}^*$ (except for some particular subclasses of Dec-POMDP, like in Dec-MDP, where the joint observation $\mathbf{o}$ forms the true underlying state $s$ itself, and thus reactive policies, i.e. policies conditioning on local observations $o_i$ rather than on local histories $\tau_i$, can still achieve the optimal solution). Is your framework specifically for this very special kind of settings, or it also works on the more general, history-based, setting? Is it is the latter, please correct the notation to align it with the existing literature, otherwise please argument such limitation.
- From Section 4.1 onward, you start writing the reward as a function of the joint observation $\mathbf{o}_i$ rather than the true state $s_t$. This is not true in general (unless in a Dec-MDP, see my point above), and you should adjust this to reflect the actual environment you have introduced earlier on.
- Where does the bias term $b(\mathbf{o})$ goes in Equation (10) and the following?
- Equation (11) is not immediate: please argument a bit more on why this is actually the case.
- One thing that immediately comes to mind in this setting is that, in order to collect a properly various and useful dataset, a very good behavioural policy is generally needed. Indeed, in the Experiments section, you say that you use a trained HAPPO policy to generate the MA-MuJoCo dataset (similarly, the SMAC dataset comes from a trained MAPPO policy). While in actual supervised learning the dataset comes from whatever source of information, and has nothing to do with the final classifier that we aim to learn, here in order to learn a policy we need another (behavioural) policy to generate these data, and the policy itself has to be good enough in order to generate a dataset with also successful and good transitions that can improve the learned policy. I would like to see a bit of discussion with respect to this point: how good has the behavioural policy to be? How much of an improvement can we gain with offline learning over that (if there is no improvement, there would be no point in learning a new policy over the behavioural one that generated the dataset)? How difficult is this kind of approach in relation to generate the required datasets? This kind of questions are unique to the offline RL setting, and does not have a counterpart in standard supervised learning, and thus I think there deserves some discussion and clarification to properly assess the merits and eventual limitations of this family of algorithms. I appreciate that section 5.5 tries to go in that direction a bit, by showing how close or far the learned policy wants to be w.r.t. the behavioural policy. This is a nice ablation study, and worth a bit more investigation in my opinion.

Typos:
- Line 54: perhaps you mean dub (rather than dumb)?
- Joint observations $\mathbf{o}$ are never introduced explicitly, although used in the work.
- Line 113: in the subscript of the expectation, shouldn't that be $\pi_{tot}$?
- Line 129: the weight is $w_i(\mathbf{o})$ alone, not the entire $w_i(\mathbf{o})Q_i(o_i,a_i)$.
- Line 177: value functions
- Line 206: optimizable (rather than "can be optimized")
- Line 212: $\pi_i^*$
- Line 229: learns
- Please expand a bit the caption of Figure 1 by describing a bit more in details what it represents.

**Limitations:**

No clear statement of possible limitations is provided in the paper.

---

> ### Author Rebuttal · Authors · 2023-08-09
>
> Thank you for your review.
>
> 1.**Policies conditioning on local observations or local histories.**   Our framework is not designed for the special setting that only considers the local observation.  It can also be applied to the action-observation history based algorithms. There is no special reason for using observations to calculate the Q-value. Although in many MARL algorithms Q-value conditions on the action-observation history $\tau$, there are still some algorithms where Q-value conditions on the observation $o$ [1] [2] [3] [4]. Thus, using the observation to calculate Q-values is not a special operation.  During the experiment, for our method OMIGA and baseline methods, we use the same input when calculating Q-values, so the effectiveness of our method is independent of whether we choose observation $o$ or history $\tau$.  We will also supplement experiments based on action-observation history in the future.
>
> 2.**The reward function based on the joint observation or the true state.**  The reward value is indeed given by the environment based on the global state. We directly use the reward  returned by the environment, which is not related to the design of the algorithm.
>
> 3.**The question of Equation(10).** What we want to express in Equation (9) is $Q_{tot}(\boldsymbol{o}, \boldsymbol{a})=\sum_i w_i(\boldsymbol{o}) Q_i(o_i, a_i)+b(\boldsymbol{o})$, $ V_{tot}(\boldsymbol{o})=\sum_i w_i(\boldsymbol{o}) V_i(o_i)+b(\boldsymbol{o})$,  $w_i \geq 0, \forall i=1 \cdots, n$,  so the global advantage should be expressed as $A_{tot}(\boldsymbol{o}, \boldsymbol{a})= Q_{tot}(\boldsymbol{o}, \boldsymbol{a})-V_{tot}(\boldsymbol{o})=\sum_i w_i(\boldsymbol{o})(Q_i(o_i, a_i)-V_i(o_i))= \sum_i w_i(\boldsymbol{o}) A_i(o_i, a_i)$. Equation (9) has a typo, and the corrected  Equation (9)  will derive  Equation (10).  Thank you for pointing out.
>
> 4.**About Equation (11).** The solution to Eq. (12) is unique because Eq. (12) is convex with respect to V. In a convex optimization problem, the feasible region -- the intersection of convex constraint functions -- is a convex region. With a convex objective and a convex feasible region, there can be only one optimal solution, which is globally optimal, that is $ V^* $ in Eq. (11).
>
> 5.**More discussion.** Thanks for pointing out that. That's a good question. In our paper, we follow those works in the offline single-agent RL setting that collected data using trained RL agents. I think that it is a general question about the offline RL setting: How good has the behavioral policy to be and how much of an improvement can we gain with offline learning. From our experience, the datasets may not need to be expert. Diversity and mediocre quality are needed for the agent to achieve a high score. For example, in hopper-medium datasets, some offline single RL methods can achieve near expert score. When the datasets’ quality is low or the diversity is limited, there may be not too much room for offline RL to improve over it. Simply using BC may achieve similar performance. Note that there are some theory papers trying to answer this question but beyond the scope of this paper. We will add some discussion about that in our revision.
>
> [1] Revisiting Some Common Practices in Cooperative Multi-Agent Reinforcement Learning, ICML 2022
>
> [2] Dealing with Non-Stationarity in MARL via Trust-Region Decomposition, ICLR 2022
>
> [3] Shared Experience Actor-Critic for Multi-Agent Reinforcement Learning, NeurIPS 2020
>
> [4] Towards Understanding Cooperative Multi-Agent Q-Learning with Value Factorization, NeurIPS 2021

---

> > ### Comment · Reviewer_WVBc · 2023-08-14
> > **Reply to Authors**
> >
> > First of all, I would like to thank the authors for their comments. Here are some additional comments:
> > 1. It is not a matter of empirical results (not only those), but rather about the theory holding. The true Q-function of the underlying multi-agent system cannot be conditioned on the joint observation $\mathbf{o}$, as this is not a Markovian signal and thus cannot be used alone to predict the next-step dynamics, unless in the above mentioned specific case of Dec-MDPs.
> > 2. Agreed, that is why I was pointing out such inconsistency.
> > 4. I agree, but where does the second term in the R.H.S. of Equation (12) comes from? It does not seem to come from Equation (11)...
> > 5. Of course such a question goes widely beyond your work alone. What I was curios to know about your work in particular is the comparison between OMIGA results and the performances of the algorithm used to generate the offline dataset trajectories: if there is little or no gain in performance over the behavioural policy used for the dataset, then I wonder what the actual improvement of applying a complex technique such as offline RL (and OMIGA in this specific case) is.  It would be good to show that OMIGA is capable of doing better of its "master", to show that it can generalize well beyond the behaviour that it used to get training examples.

---

> > > ### Author Response · Authors · 2023-08-17
> > > **Response**
> > >
> > > We thank the reviewer for the reply. Here are our response to your comments/questions.
> > >
> > > 1. Thank you for pointing out that. We acknowledge that in Dec-POMDP one should use the joint action-observation history to achieve optimality. Actually, our paper is based on the setting described in one newest SOTA offline MARL algorithm, OMAR [1], which uses local observations as the state. We will revise Section 3.1 to match the settings described in [1].
> > >
> > > 2. **but where does the second term in the R.H.S. of Equation (12) comes from?** Actually the second part is coming from "1" in Equation (11). Having the second term in Equation (12) will resulting the "1" term in Equation (11) after taking the derivation with respect to V. We are willing to give the full derivation if you have trouble calculating it.
> > >
> > > 3. A good point. We have added the datasets profile to Table 1 (see the response to xgpF) and the performance of behavioural policy can be indicated from it. We can observe that, in some datasets, all offline MARL algorithms fail to match the performance of the behavioural policy. Although it's a good point to investigate why this phenomenon happens in offline MARL, here in this paper we care more about how can we extract the best policy from offline datasets, without knowing the behavior policy. The behavior policy here is only to generate those datasets because otherwise we can have meanningful datasets. Hence, in this paper it's more reasonable to compare offline MARL algorithms to BC, which is the only way we can recover the behavior performance.
> > >
> > >
> > > [1] Plan Better Amid Conservatism: Offline Multi-Agent Reinforcement Learning with Actor Rectification. ICML 2022.

---

### Official Review · Reviewer_xgpF · 2023-06-20

**Soundness:** 1 poor
**Presentation:** 2 fair
**Contribution:** 3 good
**Rating:** 5
**Confidence:** 4

**Summary:**

This paper focuses on offline multi-agent reinforcement learning in the centralized training decentralized execution setting and proposes a SARSA-like approach that leverages value decomposition. Drawing from entropy-regularized and maxent RL, the authors propose a Behavior-Regularized formulation for Offline RL. The method is tested on offline MAMuJoCo datasets generated by the authors as well as SMAC datasets. The baselines are two single agent methods (BCQ and CQL) that the authors extend to the multi-agent setting and two multi-agent offline methods (ICQ and OMAR). Authors also investigate two ablations of their method.

**Strengths:**

- Offline multi-agent reinforcement learning is of significant interest.
- Behavior-Regularized formulation of Offline RL is interesting.
- The paper is overall easy to read and understand.

**Weaknesses:**

I believe there are major flaws with the method and the experimental procedure. I think that the method is quite similar to Behavioural Cloning (BC) and is unlikely to improve much on it or on the datasets' performance. This failed to be evidenced by the experimental results because 1) the method is not compared against BC and 2) the datasets' average and median scores are not reported. Additionally, I think that BCQ and CQL have been poorly extended into multi-agent baselines and that how results are presented in table 1 misleads the reader into believing that the presented method significantly outperforms the baselines. I detailed this point below


- The method does not improve on BC: Eq. (15) would yield pi^star, but only if Q and V are Q^star and V^star. Which is not the case since they are learned from (14) which simply learns the Q value of the behaviour policy on the dataset (there is no greedification happening here) and V is just learned to normalize the policy to one. Therefore, the only greedification step is (15) (i.e. pi is a Boltzmann policy w.r.t. to the behaviour policy values). Yet, since the learned policy does not influence any of the value learning objectives, this does not greedify the value functions. Therefore, I believe that the presented method is very close to just doing BC. For comparison, consider the greedification steps of two other SARSA-like offline RL approaches, IQL [1] and ICQ: IQL explicitly greedifies the learned values from expectile-learning before doing AWR (AWR being equivalent to (15)).  In ICQ, the greedified policy greedifies the value learning through the importance sampling correction.

    Therefore, I doubt that the presented method improves on the simple BC approach. Comparison to BC and reporting the datasets' performance is essential in order to show that the method can improve on the datasets and BC baseline. Regarding that last point, by quickly looking at the results in [2], it seems that OMIGA is not really able to improve on BC or match the average dataset performance on SMAC.

    I would also recommend comparing with a multi-agent version of IQL and independent-TD3+BC that explicitly regularizes the offline policy with BC (see for instance baselines description in [3]).

- Table 1 is misleading: bold values should denote values that are above others with statistical significance. Yet, in Table 1. Most bold values' confidence interval overlap with the ones of other values (at least for SMAC). This is quite misleading as it suggest that one method outperforms the others, while actually most methods are equivalent statistically speaking.

- BCQ and CQL have been poorly extended into multi-agent baselines: the multi-agent extensions use centralized values (with value decomposition) but decentralized regularization, which seems to be specifically what authors have shown not to work in their ablation of OMIGA. I say "seem" because it is unclear from section 5.4 whether or not values are learned with a centralized value and value decomposition.
    I believe that MA-BCQ and MA-CQL should be either fully decentralized ([4] shows that fully decentralized tends to be better for offline MARL) or with centralized regularization (like OMIGA).


EDIT: I have raised my score to 5 after the discussion period

**Questions:**

- the very poor performance of OMAR on Mujoco in table 1. is surprising as OMAR is reported to outperform MA-ICQ and MA-CQL in multi-agent continuous control [4].
- how were the hyperparameters tuned and selected for OMIGA and the baselines on each task?
- l.192 have authors investigated using different weights for V and Q?
- l.51 "some particular choice" of what?

### Summary of review
Despite offline MARL being a challenge of significant importance, and the behavior-regularized formulation proposed here being interesting, I believe that the presented method and its evaluations have too many flaws to allow for publication.

### references
[1] Kostrikov, Ilya, Ashvin Nair, and Sergey Levine. "Offline Reinforcement Learning with Implicit Q-Learning." International Conference on Learning Representations (2022).

[2] Meng, Linghui, et al. "Offline pre-trained multi-agent decision transformer: One big sequence model conquers all starcraftii tasks." arXiv preprint arXiv:2112.02845 (2021).

[3] Barde, Paul, et al. "A Model-Based Solution to the Offline Multi-Agent Reinforcement Learning Coordination Problem." arXiv preprint arXiv:2305.17198 (2023).

[4] Pan, Ling, et al. "Plan better amid conservatism: Offline multi-agent reinforcement learning with actor rectification." International Conference on Machine Learning. PMLR, 2022.

---

> ### Author Rebuttal · Authors · 2023-08-09
>
> We thank the reviewer for the comments. However, I believe some of your understanding of our paper is wrong. See the reply as follows.
>
> **The method does not improve on BC.** Our method is totally different from BC.
> - From one perspective, our overall learning objective is Equation (2), which adds an behavior regularization term to the reward. In Equation (2), hyperparameter $\alpha$ controls the strength between doing Q-learning and doing BC. If $\alpha=0$, the learning objective return backs to the original RL objective. Due to the reason that $ V^* $, $ Q^* $ and $ \pi^* $ (in Proposition 4.2) are closed-form solution of Equation (2) and $Q$, $V$ will converge to the $ Q^* $, $ V^* $ in Theorem 1, our method is definitely doing RL rather than BC.
> - From another perspective, in OMIGA, there exists one greedification step in **learning the value function V**. In Equation (13), the first term always pushes up V-values and the gradient becomes larger if Q − V is bigger than 0 while the second term always pushes down V-values, and $\alpha$ trades off these two terms. This actually doing implicit maximization over actions in the dataset support, in other words, V is fitting $\max\_{a\_i^{\prime}  \text { s.t. } \mu\_i\left(a\_i^{\prime} \mid o\_i^{\prime}\right)>0 } \bar{Q}\_i (o'\_i, a'\_i)$. Please refer to Appendix A in [1] to get a more concrete understanding about the learning objective of V.
> We also add the results of BC against OMIGA in the following table, it can be seen that OMIGA outperforms BC by a large margin.
>
> |           task            | **BC**  |       task        | **BC** |
> | :-----------------------: | :-----: | :---------------: | :----: |
> |       Hopper expert       |  96.92  |   5m_vs_6m good   |  7.13  |
> |       Hopper medium       | 514.95  |  5m_vs_6m medium  |  7.31  |
> |   Hopper medium-replay    | 192.22  |   5m_vs_6m poor   |  6.44  |
> |   Hopper medium-expert    |  90.18  |  2c_vs_64zg good  | 16.42  |
> |        Ant expert         | 2040.41 | 2c_vs_64zg medium | 14.80  |
> |        Ant medium         | 1418.06 |  2c_vs_64zg poor  | 11.12  |
> |     Ant medium-replay     | 999.88  |   6h_vs_8z good   | 11.91  |
> |     Ant medium-expert     | 1553.84 |  6h_vs_8z medium  |  9.67  |
> |    HalfCheetah expert     | 2932.75 |   6h_vs_8z poor   |  8.99  |
> |    HalfCheetah medium     | 2480.66 |   corridor good   | 10.14  |
> | HalfCheetah medium-replay | 2248.55 |  corridor medium  |  8.29  |
> | HalfCheetah medium-expert | 1846.96 |   corridor poor   |  3.44  |
>
> **The question of Table 1.** For multi-agent MuJoCo tasks, our algorithm performs significantly better than the baseline algorithms. For SMAC tasks, the environment will normalize the final return within a narrow range. Thus, in some tasks, the gap between algorithms may seem not very huge. Overall, our algorithm consistently outperforms other baseline methods.
>
> **The performance of OMAR.** Due to the different datasets used in our paper and OMAR, it is possible to obtain different results. Additionally, as OMAR is an improvement based on CQL-MA, its performance will be affected by CQL-MA. Our implementation of OMAR is based on the official code of OMAR, and our code implementation of OMAR is included in the supplementary materials.
>
> **The question of baselines.** Currently, the multi-agent versions of BCQ and CQL are still very popular offline multi-agent baselines. Papers [2] [3] [4]  all use the multi-agent versions of BCQ and CQL as baselines. They all use CTDE framework and value decomposition with local-level regularization as our baselines.  Due to the impact of partial observability, non-stationary environment and the problem of credit assignment, the fully decentralized algorithm cannot achieve good results in MARL. For most MARL algorithms, it is common to use CTDE or fully centralized framework to replace decentralized training.
>
> **The question of hyperparameters.** One advantage of the OMIGA algorithm is that only $\alpha$ is the main hyperparameter. Due to the limited number of hyperparameters in OMIGA, parameter tuning is very easy, which means hyperparameters can be selected with a simple grid search. Besides, we have tested the baseline algorithms before setting their hyperparameters, and there are no unfair hyperparameters.
>
> **The question of  the weight function.** The reason why we use the same weight function for V and Q is that the credit assignment on the global Q and V should be related. If different weight functions are used for V and Q, it is difficult to guarantee that the algorithm will learn this correlation.  Another benefit of using the same weight function for V and Q is that the formula can be simplified in the subsequent derivation process. It also avoids learning too many parameters, which affects the convergence speed of the algorithm.
>
> **The problem of l.51.** It refers to the content in Eq. (7),  Eq. (8) and section 4.2.  After getting a behavior-regularized MDP with a global-level regularization,  we use Lagrangian objective function and  Karush-Kuhn-Tucker (KKT) conditions to get Proposition 4.2 (Appendix A.2).  Next, we utilize the value decomposition and the property of the exponential function, which enables the optimal global policy $\pi^*_{tot}$ can be decomposed into a combination of optimal local policies $\pi^*_{i}$ (Proposition 4.3). It naturally bridges global value regularization and value decomposition, and converts the relationship between global policies and values to the relationship between local policies and values.
>
> [1] Offline RL with No OOD Actions: In-Sample Learning via Implicit Value Regularization, ICLR 2023.
>
> [2] Believe What You See: Implicit Constraint Approach for Offline Multi-Agent Reinforcement Learning, NeurIPS 2021
>
> [3] Offline Multi-Agent Reinforcement Learning with Coupled Value Factorization, AAMAS 2023
>
> [4] Offline pre-trained multi-agent decision transformer: One big sequence model conquers all starcraftii tasks, arXiv preprint 2021.

---

> > ### Comment · Reviewer_xgpF · 2023-08-14
> > **Thank you for you response**
> >
> > I thank the authors for the response.
> >
> > As I mentioned in my original review, could the authors please report here an updated version of Table 1. with:
> > - BC (mean and std) alongside the other baselines and OMIGA
> > - the mean, median, max and min score of the demonstrations in each task*dataset
> > - bold all the best performances for which the confidence intervals overlap. That is, for each task*dataset, bold the score with the highest mean and all the other scores for which the confidence interval overlaps with the one of the highest mean. Also, please bold the mean/median dataset's score if it is within the best confidence interval.
> >
> > This would greatly help me asses the experimental benefits of OMIGA.

---

> > > ### Author Response · Authors · 2023-08-16
> > > **Response**
> > >
> > > Thank you for your reply.
> > >
> > > We add more seeds of BC, rebold the table and also record the mean, median, max and min of value of the datasets. Attached is the updated Table 1 according to your request.
> > >
> > > The experimental benefits of OMIGA can be easily found from the table: OMIGA achieves the *highest* (i.e., the bold one) score in **23 out of 24** tasks, beating all other baselines (the best baseline, ICQ, only achieves 10 out of 24). Besides, OMIGA clearly surpasses BC, always by a large margin, showing the benefits of doing RL rather BC.
> > >
> > > To give a more thorough statistical analysis of how good OMIGA is comparing to other baselines. We adopt techniques from [1], which proposes more reliable evaluation protocols of deep RL algorithms, instead of widely-used point estimates of aggregate performance such as mean and median scores. According to [1], if one is interested in knowing how robust an improvement from an algorithm X over an algorithm Y is, one metric to consider is the average probability of improvement – this metric shows how likely it is for X to outperform Y on a randomly selected task.
> > >
> > > Formally, this paper uses the Mann-Whitney U-statistic [2], for two algorithm X and Y, running on M tasks each with $N$ seeds, the score is computed by:
> > >
> > > $ P\left(X>Y\right)=\frac{1}{MN^2} \sum\_{i=1}^M \sum\_{i=1}^N \sum\_{j=1}^N S\left(x\_{m, i}, y\_{m, j}\right) \quad \text{ where } S(x, y)=1 \text { if } y<x \text { else } 0 $.
> > >
> > > Note that if the probability of improvement is higher than 0.5, then the results are statistically significant. Furthermore, if the score is higher than a threshold of 0.75, then the results are said to be statistically meaningful.
> > >
> > > In our paper, we compare OMIGA with two strongest baselines, ICQ and MA-CQL. The results are $P(\text{OMIGA} > \text{ICQ}) = 0.775$ and $ P(\text{OMIGA} > \text{MA-CQL}) = 0.875$. These results again reveal the experimental benefits of OMIGA.
> > >
> > >
> > > [1] Deep Reinforcement Learning at the Edge of the Statistical Precipice. NeurIPS 2021.
> > >
> > > [2] On a test of whether one of two random variables is stochastically larger than the other. The annals of mathematical statistics, 1947.

---

> > > ### Author Response · Authors · 2023-08-16
> > > **Updated Table 1**
> > >
> > > |         task         | dataset(mean,median,max,min)           |       **BC**       |       BCQ-MA        |       CQL-MA        |        ICQ         |       OMAR       | **OMIGA**(ours)    |
> > > | :------------------: | -------------------------------------- | :----------------: | :-----------------: | :-----------------: | :----------------: | :--------------: | ------------------ |
> > > |   Hopper (expert)    | **2452.02**, 2584.91, 3762.68, 95.27   |   209.85±191.12    |     77.85±58.04     |    159.14±313.83    | **754.74±806.28**  |    2.36± 1.46    | **859.63±709.47**  |
> > > |       (medium)       | 723.57, 696.23, 2776.48, 128.38        |    511.95±7.43     |     44.58±20.62     |    401.27±199.88    |    501.79±14.03    |   21.34±24.90    | **1189.26±544.30** |
> > > |   (medium-replay)    | 746.42, 375.7, 2801.14, 70.75          |    133.31±53.54    |     26.53±24.04     |     31.37±15.16     |   195.39±103.61    |    3.30±3.22     | **774.18±494.27**  |
> > > |   (medium-expert)    | **1190.61**, 786.31, 3762.68, 95.27    |   155.30±111.53    |     54.31±23.66     |    64.82±123.31     | **355.44±373.86**  |    1.44±0.86     | **709.00±595.66**  |
> > > |     Ant (expert)     | 2055.07, 2054.73, 2124.15, 1994.02     |    2046.31±6.18    |   1317.73±286.28    | **1042.39±2021.65** | **2050.00±11.86**  |  312.54±297.48   | **2055.46±1.58**   |
> > > |       (medium)       | **1418.70**, 1421.33, 1473.85, 840.76  |  **1421.09±7.88**  |    1059.60±91.22    | **533.90±1766.42**  | **1412.41±10.93**  | -1710.04±1588.98 | **1418.44±5.36**   |
> > > |   (medium-replay)    | 1029.51, 977.02, 1517.05, 895.37       |    994.00±20.29    |    950.77±48.76     | **234.62±1618.28**  |   1016.68±53.51    | -2014.20±844.68  | **1105.13±88.87**  |
> > > |   (medium-expert)    | **1736.88**, 1733.94, 2124.15, 840.76, |   1561.70±64.81    |   1020.89±242.74    | **800.22±1621.52**  |   1590.18±85.61    |  -2992.80± 6.95  | **1720.33±110.63** |
> > > | HalfCheetah (expert) | 2785.10, 3428.65, 3866.08, 317.93      | **3251.22±386.83** |   2992.71±629.65    |   1189.54±1034.49   | **2955.94±459.19** |  -206.73±161.12  | **3383.61±552.67** |
> > > |       (medium)       | 1425.66, 1623.81,2113.52, 251.93       |   2280.32±178.22   | **2590.47±1110.35** |   1011.35±1016.94   |   2549.27±96.34    |  -265.68±146.98  | **3608.13±237.37** |
> > > |   (medium-replay)    | 655.76, 522.8, 2132.6, -198.76         |   1886.20±390.77   |   -333.64±152.06    | **1998.67±693.92**  | **1922.42±612.87** |  -235.42±154.89  | **2504.70±83.47**  |
> > > |   (medium-expert)    | 2105.38, 1882.61, 3866.08, 251.93      |   2451.95±782.99   | **3543.70±780.89**  |   1194.23±1081.06   |   2833.99±420.32   |  -253.84± 63.94  | 2948.46±518.89     |
> > > |   5m_vs_6m (good)    | **20.00**, 20.00, 20.00, 20.00         |     6.95±0.47      |      7.76±0.15      |      8.08±0.21      |     7.87±0.30      |    7.40±0.63     | **8.25±0.37**      |
> > > |       (medium)       | **11.03**, 11.06, 11.96, 10.08         |     7.05±0.82      |      7.58±0.10      |      7.78±0.10      |    **7.77±0.3**    |    7.08±0.51     | **7.92±0.57**      |
> > > |        (poor)        | **8.50**, 8.72, 9.89, 1.81             |     6.98±0.47      |    **7.61±0.36**    |      7.43±0.10      |     7.26±0.19      |  **7.27±0.42**   | **7.52±0.21**      |
> > > |  2c_vs_64zg (good)   | **19.94**, 20.41, 21.61,15.18          |     **17.90±1.30**     |   **19.13±0.27**    |   **18.48±0.95**    |     18.82±0.17     |    17.27±0.78    | **19.15±0.32**     |
> > > |       (medium)       | 13.00, 13.23, 15.00, 10.01             |     13.37±1.87     |     15.58±0.37      |     12.82±1.61      |   **15.57±0.61**   |    10.20±0.20    | **16.03±0.19**     |
> > > |        (poor)        | 8.89, 9.14, 10.00, 2.53                |     11.56±0.39     |     12.46±0.18      |     10.83±0.51      |     12.56±0.18     |    11.33±0.50    | **13.02±0.66**     |
> > > |   6h_vs_8z (good)    | **17.84**, 17.12, 20.02, 15.01         |     10.02±1.67     |     12.19±0.23      |     10.44±0.20      |     11.81±0.12     |    9.85±0.28     | **12.54±0.21**     |
> > > |       (medium)       | 11.96, 11.78, 14.99, 10.00             |     9.46±0.35      |     11.77±0.16      |     11.29±0.29      |     11.13±0.33     |    10.36±0.16    | **12.19±0.22**     |
> > > |        (poor)        | 9.12, 9.35, 9.99, 4.80                 |     8.57±0.81      |     10.84±0.16      |   **10.81±0.52**    |     10.55±0.10     |    10.63±0.25    | **11.31±0.19**     |
> > > |   corridor (good)    | **19.88**, 20.11, 20.49, 15.01         |     10.81±2.61     |   **15.24±1.21**    |      5.22±0.81      |   **15.54±1.12**   |    6.74±0.69     | **15.88±0.89**     |
> > > |       (medium)       | **13.07**, 13.22, 14.99, 10.02         |     7.39±0.79      |     10.82±0.92      |      7.04±0.66      |   **11.30±1.57**   |    7.26±0.71     | **11.66±1.30**     |
> > > |        (poor)        | 4.93, 4.55, 9.99, 0.0                  |     2.91±0.57      |      4.47±0.94      |      4.08±0.60      |     4.47±0.33      |    4.28±0.49     | **5.61±0.35**      |

---

> > > > ### Comment · Reviewer_xgpF · 2023-08-17
> > > >
> > > > Thank you for your reply and taking the time to provide this additional Table.
> > > >
> > > > From the provided results:
> > > > - it remains unclear to me whether the proposed method improves on the mean dataset performance (for instance, shouldn't dataset mean be bold for Hopper medium-replay?) except for half-cheetah. I agree that it is ok to match the dataset performance for expert datasets, but I would expect more significant improvements on suboptimal datasets.
> > > > - I also do not understand why is BC unable to match mean/median dataset performances.
> > > >
> > > > Regarding the derivation of the OMIGA method:
> > > > - I do not see where Equation (2) is used in Algorithm 1, so I am still unsure whether this is what OMIGA actually optimizes.
> > > > - Setting alpha to 0, as the authors suggest, in Algorithm 1 would break equations (15), (13), and (9) so I do not see OMIGA being compatible with that setting and doing RL for alpha = 0.
> > > > - I do not see how Eq. (13) is a greedification step, for a fix Q value there is a matching V that Eq. (13) retrieves. And Q is learned with SARSA in Eq. (14) (so no greedification on Q w.r.t. the dataset). Also, both terms in Eq. (13) incorporate a division by alpha so the trade-off w.r.t. alpha is unclear to me.
> > > > - The way I see Algorithm 1: Eq. (13) and (14) are a SARSA approach to learning the Advantage function of the dataset policy (no greedification) and Eq. (15) learns a policy from Advantage weighted regression (BC weighted with advantage, small greedification but that is not fed back since (13), (14) and (9) are independent of the learned policy). For me, this is very similar to BC with a boltzmann policy model (maybe the paper should be framed as multi-agent imitation learning method that leverages a boltzmann prior for BC insead as a offline MARL method). Note that I mentioned this explanation already in my initial review, and that the authors' response did not explicitly comment on it.
> > > >
> > > > Currently, I still believe that there is a major flaw with the method that makes it perform imitation learning instead of offline RL. Additionally, I do not believe that the results are compelling in showing the experimental benefits of the method, since it is unable to significantly improve on the datasets' mean performances. Therefore, while I thank the authors for the continued discussion, I cannot recommend acceptance.

---

> > > > > ### Author Response · Authors · 2023-08-20
> > > > > **Response (1/2)**
> > > > >
> > > > > We thank the reviewer for the reply.
> > > > >
> > > > > - **I would expect more significant improvements on suboptimal datasets, do not understand why is BC unable to match mean dataset performances.**
> > > > >
> > > > > First, we want to clarify that even in offline single-agent RL, algorithms are not necessarily supposed to be able to surpass the dataset mean. It depends on the size and diversity of the dataset. For example, it is well known in imitation learning, BC could fail to match the performance of expert demonstrations when the size of demonstrations is limited[1][2]. Also, in offline single-agent RL, some study reveals that the performance of offline RL algorithms drop a lot when the size or diversity decreases[3]. In our paper, we sample 1000 episodes from original SMAC datasets in [4] to formulate our offline SMAC dataset. Also, the offline MuJoCo datasets used in our paper are collected by the converged HAPPO algorithm. which is different from the SAC algorithm that is used to generate the D4RL MuJoCo datasets. Although the size of the dataset is the same, the diversity of our datasets may be inferior to D4RL MuJoCo datasets as SAC does a better job at exploration with more stochasticity. Besides, note that we are studying the offline multi-agent setting, and there exist unique challenges, e.g., partial observations, which may cause every single agent more hard to learn a good policy from the data.
> > > > >
> > > > > One concern one may have is what's the meaning of applying a complex technique such as offline RL (and OMIGA in this specific case) if there is little or no gain in performance over the behavioral policy used for the dataset. We want to highlight that in no way this makes the experiment setting in our paper trivial or nonsense.
> > > > > In the offline setting, we care more about how can we extract the best policy from offline datasets. We do not know the behavior policy so the only way we can recover the behavior performance is BC.
> > > > > Besides, our offline datasets, especially MuJoCo datasets, bring new merits to the community: how to learn from limited/non-diverse/partial-observed datasets that occur more in the offline multi-agent setting is an open problem that requires more future study.
> > > > >
> > > > > - **I do not see where Equation (2) is used in Algorithm 1, so I am still unsure whether this is what OMIGA actually optimizes.**
> > > > >
> > > > > Let me elaborate, first note that Equation (2) is equal to maximize V(o), where $ V(o)=E_\pi[r-\alpha \log (\frac{\pi}{\mu}) + \gamma V(o') ]$. Then note that $Q = r + \gamma V$, putting it into the above equation, we can get Equation (4), OMIGA is getting by solving the closed form solution of maximizing Equation (4). So clearly OMIGA is optimizing Equation (2).
> > > > >
> > > > > - **Setting alpha to 0, as the authors suggest, in Algorithm 1 would break equations (15), (13), and (9) so I do not see OMIGA being compatible with that setting and doing RL for alpha = 0.**
> > > > >
> > > > > Sorry for the misunderstanding here. Yes, setting $\alpha = 0$ will break equations in OMIGA because $\alpha = 0$ means there's no behavior regularization, so OMIGA will also not be valid. Here we only want to clarify that OMIGA can be turning more towards doing RL by setting alpha smaller, $\alpha = 0$ is an extreme case.
> > > > >
> > > > > - **I do not see how Eq. (13) is a greedification step, for a fixed Q value there is a matching V that Eq. (13) retrieves. Also, both terms in Eq. (13) incorporate a division by alpha so the trade-off w.r.t. alpha is unclear to me.**
> > > > >
> > > > > Eq. (13) is definitely a greedification step. First, remember that Eq. (13) is optimizing V. There are two terms in Eq. (13), the second term will push down all V values while the first term will push up those V values when $Q(o, a) - V(s)$ is very large. Suppose that for a certain state, there are multiple behavior actions in the datasets (it still holds when the state won't be observed again, i.e, one behavior action per state, please see discussion in IQL for more details), V can then be fitted towards the (dataset) action that has the largest Q value by adjusting the hyperparameter $\alpha$. In other words, as we previously said, V is fitting towards the extreme value, $\max _{a \text { s.t. } \mu (a \mid o)>0} Q(o, a)$. $\alpha$ can do the trade-off because when we compute the **ratio** of the derivation of the first term w.r.t V and the second term w.r.t V, we will get $\text{exp}(\frac{w}{\alpha}(Q-V))$, which can be adjusted by $\alpha$.
> > > > >
> > > > > Another justification is that the learning objective (Equation (2), with f being $\log$) of OMIGA is a generalized version of Maximum Entropy RL, which means $V(o)=\alpha \log \sum_{a} \mu(a | o) \exp (Q(o, a) / \alpha)$ [5]. This relationship clearly shows how V is doing greedification and how $\alpha$ does the trade-off.

---

> > > > > > ### Author Response · Authors · 2023-08-20
> > > > > > **Response (2/2)**
> > > > > >
> > > > > > - **For me, this is very similar to BC with a boltzmann policy model (maybe the paper should be framed as multi-agent imitation learning method that leverages a boltzmann prior for BC insead as a offline MARL method).**
> > > > > >
> > > > > > We respectively disagree with it again. As both in the previous response and current response, we have shown multiple times the difference between OMIGA and imitation learning. Also, as far as we know, no other reviewers raised questions/concerns about it. We sincerely hope that the reviewer could in-depth read our response, rebuild the current understanding of our paper and re-evaluate our paper accordingly.
> > > > > >
> > > > > > -----
> > > > > >
> > > > > >
> > > > > > [1] Imitation Learning via Off-Policy Distribution Matching, ICLR 2020.
> > > > > >
> > > > > > [2] Discriminator-Guided Model-Based Offline Imitation Learning, CORL 2022.
> > > > > >
> > > > > > [3] Look Beneath the Surface: Exploiting Fundamental Symmetry for Sample-Efficient Offline RL, Arxiv 2023.
> > > > > >
> > > > > > [4] Offline pre-trained multi-agent decision transformer: One big sequence model conquers all starcraftii tasks, Arxiv 2023.
> > > > > >
> > > > > > [5] Extreme Q-Learning: MaxEnt RL without Entropy, ICLR 2023.
> > > > > >
> > > > > > [6] MADIFF: Offline Multi-agent Learning with Diffusion Models, Arxiv 2023.
> > > > > >
> > > > > > [7] A Model-Based Solution to the Offline Multi-Agent Reinforcement Learning Coordination Problem, Arxiv 2023.

---

> > > > > > ### Comment · Reviewer_xgpF · 2023-08-21
> > > > > >
> > > > > > I believe there is a confusion here:
> > > > > > - Eq. (2) expectation is taken with respect to data generated by the current policy (the one being max over), therefore the max over $\pi$ will also influence how we collect transitions on which the expectation is computed. This cannot be the case for OMIGA as it only deals with expectations computed with the data from the dataset. Therefore, comparisons between OMIGA (or any offline RL algorithm) and online RL break.
> > > > > > - There is no maximization in (3) and (4), and this is policy evaluation (as the authors note l. 139) and not policy optimization (i.e. no greedification). Note that in (3) and (4) V and Q should be indexed by $\pi$, the policy that generates the transitions. Therefore, there is no direct link between (2) and (4) except for $V^{\pi^*}$ where $\pi^*$ is the argmax of (2).
> > > > > > - Where is it that OMIGA is doing maximization of (4)? Where is the close form solution of max (4) ?
> > > > > > - Eq. (11) is true for the Q and V of any Boltzmann policy and relates to how Q and V are computed for boltzmann policy (V is simply the expectation of Q over actions independently of whether they are the Q and V of the optimal policy or not). So (12) is just finding the V corresponding to the Q function, regardless of whether it is the one of the optimal policy or not (V could also simply be computed as the expectation of Q over actions).
> > > > > > - So no, OMIGA is not solving (2) which would require computing the expectation for trajectories collected with the optimized policy (some Offline RL approached do this with importance sampling for instance) rather than with the trajectories in the dataset.
> > > > > > - What authors describe w.r.t. Eq. (13) is exactly a bolzmann policy on the Q-value with temperature alpha: select actions with probability equal to a softmax on the Q-values.
> > > > > > - Another way of looking at this confusion: apart from the modelling choice of enforcing that $\pi$ is modeled as a boltzmann policy, what would be the difference between OMIGA and (multi-agent) IQL [1] with the expectile set to 0.5 (no greedification) ?
> > > > > >
> > > > > > I appreciate the authors' engagement in defending their paper, yet I have spent significant time discussing with the authors and I still believe that the presented method is not sound (actually I am more and more convinced of the flaw as discussion progresses). Until I am convinced otherwise, I will argue for rejection. Also, as the authors suggest, I will not mind if other reviewers would jump in to clarify where the method is doing policy optimization instead of policy evaluation on the dataset and pinpoint if and where I am mistaken in the derivations and my understanding of Algorithm 1. To the authors, please highlight and explain what is wrong with my derivations and understanding above (specifically, what ensures that we retrieve $\pi^*$ instead of the dataset $\pi$?).
> > > > > >
> > > > > >
> > > > > > [1] Kostrikov, Ilya, Ashvin Nair, and Sergey Levine. "Offline reinforcement learning with implicit q-learning." arXiv preprint arXiv:2110.06169 (2021).

---

> > > > > > > ### Author Response · Authors · 2023-08-21
> > > > > > > **Author's response**
> > > > > > >
> > > > > > > Yes it's true that Eq. (2) expectation is taken with respect to data generated by the current policy. However, as we said again and again and again. Eq. (2) is the same as maximizing Equation (4), and OMIGA is the closed form-solution of maximizing Equation (4). **The behavior regularization term $f(\pi, \mu)$ could transfer the greedy max from policy $\pi$ to a softened max over behavior policy $\mu$ (see proofs in Appendix A for why). So that's the reason why OMIGA could use only dataset actions but get the same solution of Eq. (2)!**
> > > > > > >
> > > > > > > XQL [2] can be offline if settting $\mu$ to be the behavior policy, so we are not compare with online RL algorithms...
> > > > > > >
> > > > > > > OMIGA is doing maximization of (4) becase we want to find the optimal policy with the largest V.
> > > > > > >
> > > > > > > The close form solution of maximizing (4) is shown in Proposition 4.2... and the proof is in Appendix A.
> > > > > > >
> > > > > > > Why OMIGA is solving Equation (2)?
> > > > > > >
> > > > > > > Because
> > > > > > >
> > > > > > > 1) Equation (2) is equal to maximizing Equation (4);
> > > > > > >
> > > > > > > 2) The optimal Q and V is the Q and V value of the optimal $\pi$ that maximizes Equation (4);
> > > > > > >
> > > > > > > 3) OMIGA(Proposition 4.2) is the closed form solution of the optimal Q, V (and $\pi$).
> > > > > > >
> > > > > > > So OMIGA is solving Equation (2).
> > > > > > >
> > > > > > > We sincerely hope that the reviewer could take a look at two recent offline RL papers, which are both selected as **oral** at ICLR 2023. If the reviewer still believe our method is not sound, having big flaws, then these two papers are the same.
> > > > > > >
> > > > > > > [1] Offline RL with No OOD Actions: In-Sample Learning via Implicit Value Regularization, ICLR 2023.
> > > > > > >
> > > > > > > [2] Extreme Q-Learning: MaxEnt RL without Entropy, ICLR 2023.

---

### Official Review · Reviewer_LQB9 · 2023-07-05

**Soundness:** 2 fair
**Presentation:** 3 good
**Contribution:** 3 good
**Rating:** 6
**Confidence:** 4

**Summary:**

The paper proposed a novel offline MARL algorithm with named OMIGA. OMIGA transfers the global entropy regularization into local regularizations implicitly under the linear weighted value decomposition assumption. The empirical results on multi-agent MuJoCo and SMAC suggest that OMIGA almost outperforms current offline multi-agent SOTA baselines.

**Strengths:**

Originality: OMIGA provides a novel principled framework to decompose global regularization into equivalent implicit local regularizations for offline MARL.
Quality: The theoretical analyze supports the conversion of the global regulation and the empirical results shows that OMIGA outperforms current SOTA baselines under specific hyper-parameters.
Clarity: The paper is clearly written and well organized.
Significance: OMIGA almost outperforms current offline multi-agent SOTA baselines in the experiments.


**Weaknesses:**

The hyper-parameters seem unfair. For example, in the ICQ paper, the performance of ICQ is significantly improved when the Lagrangian coefficient changes from 10 to 100 for SMAC tasks, while this paper chooses 10 for ICQ.
Mirror: The variables $a_t$, $s_t$ in the RHS of Eq. (4) should be $a$, $s$ (below line 141).


**Questions:**

1. Could you explain why you choose the hyper-parameter for the baselines as in the paper? The hyper-parameters seem different in their origin papers.
2. Do you try the dataset collected from other MARL algorithms (other than HAPPO, MAPPO)? What is the performance of OMIGA?

---

> ### Author Rebuttal · Authors · 2023-08-09
>
> Thank you for your review.
>
> 1.**The choice of hyper-parameter for baseline.**  The SMAC datasets used in the ICQ paper are generated by the authors, and the SMAC datasets in our paper come from publicly available datasets [1], which is the largest open offline dataset on SMAC.  These two datasets were collected by different algorithms (The datasets we use are collected by MAPPO, while the datasets for ICQ are collected by DOP), and the quality of these two datasets also varies  (The datasets we use include 3 different qualities, while the datasets for ICQ are mixing datasets). Due to the different datasets we use,  there are certain differences in the selection of hyperparameters.
>
> In the ICQ paper, the Lagrangian coefficient for SMAC is set to 100 or 1000, and in our paper it is set to 10.  We also supplement the experimental results  of one seed for Lagrangian coefficient = 100 and Lagrangian coefficient = 1000 on the SMAC tasks.
>
> | task       | dataset | ICQ(1000) | **ICQ(100)** |  ICQ(10)   | **OMIGA**  |
> | ---------- | :-----: | :-------: | :----------: | :--------: | :--------: |
> | 5m_vs_6m   |  good   |   7.46    |     7.62     | 7.87±0.30  | 8.25±0.37  |
> | 5m_vs_6m   | medium  |   7.45    |     7.70     |  7.77±0.3  | 7.92±0.57  |
> | 5m_vs_6m   |  poor   |   7.43    |     7.45     | 7.26±0.19  | 7.52±0.21  |
> | 2c_vs_64zg |  good   |   17.99   |    19.30     | 18.82±0.17 | 19.15±0.32 |
> | 2c_vs_64zg | medium  |   14.89   |    16.01     | 15.57±0.61 | 16.03±0.19 |
> | 2c_vs_64zg |  poor   |   11.05   |    10.94     | 12.56±0.18 | 13.02±0.66 |
> | 6h_vs_8z   |  good   |   11.69   |    11.12     | 11.81±0.12 | 12.54±0.21 |
> | 6h_vs_8z   | medium  |   11.24   |    11.01     | 11.13±0.33 | 12.19±0.22 |
> | 6h_vs_8z   |  poor   |   9.99    |    10.25     | 10.55±0.10 | 11.31±0.19 |
> | corridor   |  good   |   15.94   |    17.06     | 15.54±1.12 | 15.88±0.89 |
> | corridor   | medium  |   10.66   |    10.37     | 11.30±1.57 | 11.66±1.30 |
> | corridor   |  poor   |   3.27    |     2.60     | 4.47±0.33  | 5.61±0.35  |
>
> It can be seen that a higher Lagrangian coefficient does not bring significant improvement to ICQ algorithm. In many tasks, a higher Lagrangian coefficient leads to a decrease in ICQ performance.
>
> In fact, we have tested the baseline algorithms before setting their hyperparameters, and there are no unfair hyperparameters.
>
> 2.**The dataset collected from other MARL.**  Currently, we have only conducted experiments on the datasets in the paper. Both HAPPO and MAPPO are widely recognized MARL algorithms. The datasets that we use contain a variety of quality data, and also contain the data from training processes (such as the medium-replay dataset). Thus, the datasets we use in this paper are convincing. We will supplement experiments on other datasets in the future.
>
> [1] Offline pre-trained multi-agent decision transformer: One big sequence model conquers all starcraftii tasks, arXiv preprint 2021.

---

> > ### Comment · Reviewer_LQB9 · 2023-08-22
> >
> > Thanks for the response. I still keep my score.

---

### Official Review · Reviewer_trCY · 2023-07-27

**Soundness:** 3 good
**Presentation:** 3 good
**Contribution:** 3 good
**Rating:** 7
**Confidence:** 4

**Summary:**

This work proposes OMIGA, a new offline multi-agent reinforcement learning (MARL) algorithm with implicit global-to-local value regularization. The policy regularization from offline reinforcement learning (RL), for avoiding distribution shifts between learned policy and behavioral policy, is first brought to centralized training in MARL. Then, the global value functions and policies are decomposed into local agent-wise counterparts. Finally, using the self-normalization constraint of local policies, the objective functions for learning local value functions are derived, which leads to the OMIGA. The proposed method shows good performance compared to relevant existing works in offline single-agent and multi-agent RL, on MuJoCo and SMAC benchmark tasks.

**Strengths:**

- This work seamlessly combined the main idea/challenges from offline RL (i.e. regularization for tackling distribution shift) and MARL (i.e. value decomposition for tackling scalability). Particularly, I deem incorporating global information into local value/policy learning as a crucial matter in the field of MARL in general. In this context, the stance of this paper is well-positioned.

- The motivations and the significance of each step of deriving the OMIGA algorithm were addressed well.

- Ablation studies appropriately tackle the main sources of design choices that could occur within OMIGA: the incorporation of global observation $\boldsymbol{o}$ and the regularization parameter $\alpha$.


**Weaknesses:**

While it was difficult to find a critical weakness of the proposed algorithm within the scope of this paper, I have numerous questions regarding the derivation of the OMIGA algorithm as well as the presented results. These questions are listed in the "Questions" section.

Meanwhile, I outline below some of the confusing typos or slight inconsistencies that might confuse the readers:

- Notation mismatch between the joint policy $\pi_{tot}$ (L112) and $\boldsymbol\pi$ (L113)

- L136: It seems that the “derivation of $\pi$ and $\mu$” should actually be the “deviation between $\pi$ and $\mu$”.

- The operator $\mathcal T_f^*$ has been defined for the action-value function $Q$ in Eq. (5) but is applied on state-value functions $V_{tot}$’s throughout Appendix Section A.1 and in the equation under L492 (Appendix). While its definition is obvious in Section A.1, it would be appreciated if you could explicitly provide the definition.

- The $\beta(a|s)$ in L487 (Appendix A.2) should be $\beta(\boldsymbol{a}|\boldsymbol{o})$.


**Questions:**

I have listed below my questions regarding my confusion regarding the derivations or missing details in the experiments.

- Could you explain the derivation of Eq. (5)? Substituting $f(\pi_{tot}, \mu_{tot}) = \log (\pi_{tot} / \mu_{tot})$ into the equation under L153 yields
  - $r({\boldsymbol{o}}_0, {\boldsymbol{a}}_0) - \alpha \log(\pi_{tot}({\boldsymbol{a}}_0 | {\boldsymbol{o}}_0) / \mu_{tot}({\boldsymbol{a}}_0 | {\boldsymbol{o}}_0)) + \sum_{t=1}^{\infty} (\cdots)$,
  - but I am not sure if Eq.(5) would cover the $-\alpha \log(\pi_{tot}({\boldsymbol{a}}_0 | {\boldsymbol{o}}_0) / \mu_{tot}({\boldsymbol{a}}_0 | {\boldsymbol{o}}_0))$ term.

- In the derivation of Eq. (10), it seems that the $b(\boldsymbol{o})$ term from Eq. (9) is omitted for simplicity. Will this leave the rest of the exposition unchanged, particularly during the derivation of Eq. (11) and (12)?

- In L222-223, it is mentioned that "The proof [of Proposition 4.4] follows [from the fact] that the first-order optimality condition of the above optimization objective (i.e., derivative with respect to $V_i$ equals 0) is exactly the condition Eq. (11)." Yet I could not comprehend how the solution to Eq. (11) is unique. In other words, it would be appreciated if the authors could explain how satisfying Eq. (11) will necessarily mean that the solution of Eq. (12) is $V_i^*$.

- In the “Discussion with prior works” paragraph (L247-265), OMIGA is compared to ICQ, DMAC, and IVR. Among these, only ICQ is tested in the experiments. If feasible within the rebuttal period, could you also provide the experimental results of DMAC and IVR?

- In L262-265, it is mentioned that propagating regularization in a global-to-local direction "justifies why previous offline multi-agent methods that apply local behavior regularization can work." Could you provide an example of this insight on existing works, such as ICQ to which OMIGA is compared experimentally?

- The pseudocode of OMIGA (Algorithm 1) uses a target network counterpart $\bar Q_i$ for local action-value function $Q_i$, rather than for local state-value function $V_i$. Considering that target networks are often used for calculating targets in the critic loss, and that this target is $r(\boldsymbol o, \boldsymbol a) + \gamma V_{tot}(\boldsymbol o')$ in Eq. (14), it seems that the target networks should be kept for state-value functions instead.

- The overall scales of the scores (i.e. hundreds to thousands) in the MuJoCo tasks for Table 1 do not seem to match the scales of the reward distribution (i.e. ones) of the datasets outlined in Table 2 (Appendix B). Could you explain this discrepancy?

- How long were the OMIGA and the baseline methods trained for? Could you also provide the learning curves for the main results?

- How large are the datasets for MuJoCo?

- It seems from Table 6 (Appendix) that despite the statement that "on these mixed datasets, the behavior policy is suboptimal and more heterogeneous" (L300-301), all the methods achieve surprisingly good stability in terms of variance. Could you explain how this could occur not only for OMIGA but also for the other baseline methods?

- Will the offline datasets for MuJoCo be publicly released? This would be an immense contribution to the offline MARL community.


**Limitations:**

No limitations or potential negative societal impacts have been discussed in the submission. A limitation of the proposed work is that the underlying global value functions $Q_{tot}^*$ and $V_{tot}^*$ are assumed to be a linear combination of the local value functions $Q_i^*$ and $V_i^*$ in Eq. (9). Future work may investigate the importance of extending to nonlinear combination of local value functions, such as those satisfying monotonicity constraints [1] or IGM conditions [2,3].

[1] Rashid, Tabish, et al. "Monotonic value function factorization for deep multi-agent reinforcement learning." The Journal of Machine Learning Research 21.1 (2020): 7234-7284.

[2] Son, Kyunghwan, et al. "Qtran: Learning to factorize with transformation for cooperative multi-agent reinforcement learning." International conference on machine learning. PMLR, 2019.

[3] Wang, Jianhao, et al. "Qplex: Duplex dueling multi-agent q-learning." arXiv preprint arXiv:2008.01062 (2020).

---

> ### Author Rebuttal · Authors · 2023-08-09
>
> Thank you for your review. We appreciate for minor writing typos you posted, we will correct those typos in our revision.
>
> 1. **The derivation of Eq. (5).** The term, $ -\alpha \log( \frac{\pi\_{tot}({\boldsymbol{a}}_0 | {\boldsymbol{o}}_0)} {\mu\_{tot} ({\boldsymbol{a}}_0 | {\boldsymbol{o}}_0)} ) $ is contained in the value function V (see Equation (6)). Note that our derivation is almost the same as SAC, see Equation (2) and (3) in [1]. Actually, if we put Equation (5) into Equation (6), we can get $ V(o) = E\_{\pi} \left[ r - \alpha \log ( \frac{\pi}{\mu} )) + V(o') \right] $. When optimizing this "modified" value function, we are actually solving the origin MDP with a modified reward $ \hat{r} = r - \alpha \log (\pi / \mu) $, which is exactly the Equation in line 153.
>
> 2. **The derivation of Equation (10).**  We apologize that we make a typo in Equation 9, there should not have b(0) in it. $Q_{tot}(\boldsymbol{o}, \boldsymbol{a})=\sum_i w_i(\boldsymbol{o}) Q_i(o_i, a_i)+b(\boldsymbol{o})$, $ V_{tot}(\boldsymbol{o})=\sum_i w_i(\boldsymbol{o}) V_i(o_i)+b(\boldsymbol{o})$,  $w_i \geq 0, \forall i=1 \cdots, n$, so the global advantage should be expressed as $A_{tot}(\boldsymbol{o}, \boldsymbol{a})= Q_{tot}(\boldsymbol{o}, \boldsymbol{a})-V_{tot}(\boldsymbol{o})=\sum_i w_i(\boldsymbol{o})(Q_i(o_i, a_i)-V_i(o_i))= \sum_i w_i(\boldsymbol{o}) A_i(o_i, a_i)$. Note that this typo does not affect the subsequent derivation, thank you for pointing out that.
>
> 3. **The proof [of Proposition 4.4].** First we apologize for a typo in Equation (12): $Q$ should be $ Q^* $. The solution to Eq. (12) is unique because Eq. (12) is convex with respect to V.
> In a convex optimization problem, the feasible region -- the intersection of convex constraint functions -- is a convex region. With a convex objective and a convex feasible region, there can be only one optimal solution, which is globally optimal, that is $ V^* $ in Eq. (11).
>
> 4. **The experiments of DMAC and IVR.** Although we discuss the relationship between our method and DMAC and IVR, it is not necessary to add the experiments of DMAC and IVR. DMAC is an online MARL algorithm, and IVR is a single-agent offline RL algorithm. Neither DMAC nor IVR  can be directly applied to our offline multi-agent experiments. Our discussion with DMAC and IVR is only a theoretical explanation of the correlation between our method and these algorithms.
>
> 5. **Example of this insight on existing works.** Some previous work, like MA-BCQ, MA-CQL, OMAR, apply local regularization per agent and achieve good results. However, this is a kind of ad-hoc choice by reusing techniques from offline single-agent RL to multi-agent setting, there's no guarantee that doing so is correctly applying global regularization. Our work starts from the correct thing, aka, global regularization and shows it could be decomposed into local ones under certain cases.
>
> 6. **About the target network** Yes, target networks are often used for calculating targets in the critic loss. In our algorithm, state value function V are actually playing the role of Q target. However, note that V is learned by optimizing Equation (13), so we use Q target in Equation (13). In other words, we want to V to approximate $\max\_{a\_i^{\prime}  \text { s.t. } \mu\_i\left(a\_i^{\prime} \mid o\_i^{\prime}\right)>0 } \bar{Q}\_i (o'\_i, a'\_i)$.
>
> 7. **The scales of the scores.**  Table 2 (Appendix B) shows the distribution of one-step rewards for each task. However, Table 1 shows the average returns for each task. The return is calculated from one episode and contains many timesteps.
>
> 8. **Training time.** For multi-agent MuJoCo tasks, we train all methods for 200 million timesteps. For SMAC tasks, we train all methods for 50000 episodes. We will provide the learning curves in the subsequent versions of the paper.
>
> 9. **The size of multi-agent MuJoCo datasets.**  Each task includes 1 million timesteps data.
>
> 10.  **The good stability in terms of variance.**  It has a certain relationship with the experimental environment. In SMAC,  the environment will normalize the final return within a narrow range. Thus,  sometimes the final performance of the algorithm is relatively close, and the variance appears low. However, our algorithm still outperforms other baseline methods.
>
> 11. **The question of whether to release the MuJoCo dataset.** Yes, we will publicly  release our  offline datasets and open source our code.
>
> [1] Soft Actor-Critic Algorithms and Applications, arXiv preprint 2018.

---

> > ### Comment · Reviewer_trCY · 2023-08-14
> > **Thank you for your response!**
> >
> > Thank you for your response. My major concerns were with regards to the derivation of OMIGA, and these have been addressed well. As long as further revision fixes the typos in Q2, Q3 (for the typos, merely fixing them in the manuscript should be sufficient) and adds short notes regarding Q1, Q4, Q6, Q8 and Q9, I believe that this is a technically solid paper. In addition, it remains that this paper tackles the important challenge of offline MARL (i.e. decomposing global behavioural regularization to local regularization), which renders this paper significant in the field.
> >
> > I carefully read all reviews from other reviewers. While it does seem that, as another reviewer has criticized, OMIGA’s performance overlaps with those of some baselines (e.g. OMIGA vs ICQ for Hopper and Ant, expert dataset in Table 1), OMIGA still seems overall robust with respect to dataset qualities even in those cases. I believe the experimental results still strongly favour OMIGA.
> >
> > I will raise my confidence, soundness, and overall rating of this submission.

---

> > > ### Author Response · Authors · 2023-08-14
> > > **Thank you for your detailed review**
> > >
> > > Dear reviewer trCY,
> > >
> > > Thank you so much for raising your score. We really appreciate it! Your detailed review helps a lot to improve the quality and clarity of our paper. We will fix the typos and revise our paper in the final version to be more clear.
> > >
> > > Thanks!
> > >
> > > Best regards,
> > >
> > > Authors of Paper 5814

---

### Decision · Program_Chairs · 2023-09-21

**Decision:**

Accept (poster)

**Comment:**

The authors present OMIGA, a new offline multi-agent reinforcement learning (MARL) algorithm with implicit global-to-local value regularization. This work combines the main idea/challenges from offline RL of regularization for tackling distribution shift and MARL of using value decomposition for tackling scalability. Reviewers found the paper clear and algorithm well-motivated. Specifically, the motivations and the significance of each step of deriving the OMIGA algorithm were addressed and ablation studies appropriately tackle the main sources of design choices that could occur within OMIGA. There was much discussion among reviewers about optimality of the algorithm due to its closeness to BC, and acknowledgement of issues due to this. However, the derivation remains a good inspiration for devising the final algorithm OMIGA, and to address the gap between Eq. (12) and Eq. (13), the authors have provided explanations for "greedification" within Eq. (13), which a reviewer found intuitive, and this gap does not seem to be significant enough to affect the experimental results.

Due to the strength of the empirical results I vote for acceptance. However, I do believe that Reviewer xgpF's concerns are valid, and a discussion of the limitations of the derivation of the final algorithm necessary for the camera ready.